# SPIKEPINGPONG: SPIKE VISION-BASED FAST-SLOW PINGPONG ROBOT SYSTEM

**Hao Wang**[1,2,*]**, Chengkai Hou**[1,*]**, Xianglong Li**[2,*]
**Yankai Fu**[1]**, Chenxuan Li**[1]**, Ning Chen**[1]**, Gaole Dai**[1]**, Jiaming Liu**[1]
**Tiejun Huang**[1,2]**, Shanghang Zhang**[1,2,†]
[1]State Key Laboratory of Multimedia Information Processing,
School of Computer Science, Peking University
[2]Beijing Academy of Artificial Intelligence (BAAI)
`shanghang@pku.edu.cn`
 * Equal contribution    † Corresponding author

## ABSTRACT

Learning to control high-speed objects in dynamic environments represents a fundamental challenge in robotics. Table tennis serves as an ideal testbed for advancing robotic capabilities in dynamic environments. This task presents two fundamental challenges: it requires a high-precision vision system capable of accurately predicting ball trajectories under complex dynamics, and it necessitates intelligent control strategies to ensure precise ball striking to target regions. High-speed object manipulation typically demands advanced visual perception hardware capable of capturing rapid motion with exceptional temporal resolution. Drawing inspiration from Kahneman's dual-system theory, where fast intuitive processing complements slower deliberate reasoning, there exists an opportunity to develop more robust perception architectures that can handle high-speed dynamics while maintaining accuracy. To this end, we present *SpikePingpong*, a novel system that integrates spike-based vision with imitation learning for high-precision robotic table tennis. We develop a Fast-Slow system architecture where System 1 provides rapid ball detection and preliminary trajectory prediction with millisecond-level responses, while System 2 employs spike-oriented neural calibration for precise hittable position corrections. For strategic ball striking, we introduce Imitation-based Motion Planning And Control Technology, which learns optimal robotic arm striking policies through demonstration-based learning. Experimental results demonstrate that *SpikePingpong* achieves a remarkable 92% success rate for 30 cm accuracy zones and 70% in the more challenging 20 cm precision targeting. This work demonstrates the potential of Fast-Slow architectures for advancing robotic capabilities in time-critical manipulation tasks.

## 1 INTRODUCTION

Current research in robot learning primarily focuses on manipulation tasks involving static or slow-moving objects (Avigal et al., 2022; Wu et al., 2024; Wang et al., 2024; Luo et al., 2024; Vuong et al., 2024; Shao et al., 2020). While these achievements represent significant progress, they predominantly address scenarios with relatively simple dynamics and predictable object behaviors. However, real-world environments are replete with dynamic scenarios involving high-speed moving objects that demand rapid perception and precise control, from catching falling items (Zhang et al., 2024b) and intercepting projectiles (Natarajan et al., 2024) to navigating through crowded environments (Gao et al., 2023). These high-speed scenarios present fundamentally more challenging problems requiring millisecond-level decision making and robust handling of dynamic uncertainties.

Table tennis provides an ideal testbed for developing such capabilities, as it constitutes an optimal paradigm for high-speed robotic interaction while exhibiting exceptional generalizability. This task embodies Moravec's paradox (Moravec, 1988) in its purest form: what appears as a simple recreational activity to humans represents one of the most challenging domains for robotics, demanding

Figure 1: **Overview of *SpikePingpong*.** The framework integrates two key stages: (1) Interception, using a Fast-Slow architecture for precise trajectory prediction, and (2) Striking, employing the IMPACT module to execute strategic returns via imitation learning. The system achieves a 92% overall success rate and 70% in high-precision targeting tasks.

the integration of high-speed perception, predictive modeling, and precise motor control. Beyond its apparent simplicity, this task systematically encapsulates the fundamental challenges of dynamic robotics: millisecond-scale perception and prediction, precise manipulation under temporal constraints, and real-time strategic planning. The core competencies developed, including high-speed object tracking, precision manipulation, and adaptive control, demonstrate direct transferability to industrial automation Deka et al. (2024), medical robotics Wah (2025), and aerospace trajectory interception systems Baradaran (2025). This inherent scalability positions table tennis as a systematic methodology for developing foundational capabilities essential for advanced robots operating in dynamic, time-critical environments.

When tackling high-speed dynamic tasks like robotic table tennis, existing approaches can be broadly categorized into control-based methods and learning-based methods. Control-based approaches (Acosta et al., 2003; Mülling et al., 2010; Zhang et al., 2018; Silva et al., 2005; Yang et al., 2010) rely on precise physical modeling and predefined motion planning, but despite being mathematically rigorous and computationally efficient, they struggle with real-world complexities due to their requirement for precise calibration and inability to adaptively adjust to varying ball trajectories or unexpected disturbances. Learning-based approaches (Mülling et al., 2013; Abeyruwan et al., 2023; D'Ambrosio et al., 2023; Gao et al., 2020; Ding et al., 2022; Zhao et al., 2024; DAmbrosio et al., 2025; Su et al., 2025), offer greater theoretical adaptability. However, they often suffer from the persistent sim-to-real gap, where policies trained in simulation perform poorly in physical systems. This is particularly pronounced in table tennis, where subtle factors like ball spin and contact dynamics significantly impact performance. Additionally, existing methods typically rely on high-precision hardware vision systems, which are expensive and may still struggle with the rapid temporal dynamics required for accurate trajectory prediction. Drawing inspiration from Kahneman's dual-system theory (Kahneman, 2011), where fast intuitive processing (System 1) complements slower deliberate reasoning (System 2), there exists an opportunity to develop more robust perception architectures that can handle high-speed dynamics while maintaining accuracy.

To this end, we propose *SpikePingpong*, a high-precision robotic table tennis system integrating spike vision-based Fast-Slow system and advanced control techniques as illustrated in Figure 1. Our system addresses the fundamental challenge of robotic table tennis through a principled decomposition into interception and striking phases, each employing specialized technical innovations. **For the interception phase,** we employ a Fast-Slow system architecture where System 1 provides rapid ball detection and preliminary trajectory prediction with millisecond-level responses, while System 2 leverages high-frequency spike camera data for refined trajectory analysis through neural error correction, effectively addressing physical model inaccuracies caused by environmental variables and spin dynamics. **For the striking phase,** we develop Imitation-based Motion Planning And Control Technology (IMPACT), which learns strategic ball striking through demonstration-based learning, mapping incoming trajectory characteristics to optimal robotic arm striking policies. In summary, our contributions are as follows:

- We design and implement a comprehensive robotic table tennis system that systematically addresses high-speed dynamic manipulation through task-specific decomposition and Fast-Slow architecture.

- We develop a Fast-Slow system perception framework that enables accurate trajectory prediction using conventional cameras through neural error correction, complemented by real-world imitation learning for precise ball striking control.

- We conduct extensive experimental evaluation demonstrating superior performance with 92% success rate in 30cm zones and 70% accuracy in challenging 20cm precision targeting, validating the effectiveness of our integrated approach.

## 2 RELATED WORK

### 2.1 AGILE POLICY LEARNING

Agile policy learning addresses the challenge of generating fast, adaptive, and robust behaviors in highly dynamic environments. It has been widely studied across various domains such as autonomous driving (Pan et al., 2017; Pomerleau, 1988; Muller et al., 2005; Anzalone et al., 2022; Pan et al., 2020), legged locomotion (Nguyen et al., 2017; Tan et al., 2018; Haarnoja et al., 2018; Zhong et al., 2025), humanoid skills (He et al., 2025; Ben et al., 2025; He et al., 2024; Zhang et al., 2024a), and dynamic manipulation tasks like throwing and catching (Zhang et al., 2024b; Hu et al., 2023; Kim et al., 2014; Huang et al., 2023; Lan et al., 2023). These tasks require policies capable of maintaining high inference frequencies, handling disturbances, and generalizing across a wide range of conditions. Perception plays a critical role in enabling robots to adapt to environmental changes (Wang & Wang, 2022) and to understand the dynamic interaction between objects and agents (Zeng et al., 2020; Kober et al., 2011; Fu et al., 2025). When tracking fast-moving objects, systems often rely on high-speed motion capture setups (Mori et al., 2019). However, in table tennis scenarios, the high speed and abrupt motion of the ball cause significant motion blur with standard RGB cameras, leading to inaccurate position estimates and trajectory predictions. Unlike previous systems that rely on high-precision hardware setups, our *SpikePingpong* system introduces a Fast-Slow system architecture that integrates innovative spike-based vision technology with imitation learning. The Fast system provides rapid ball detection and preliminary trajectory estimation, while the Slow system leverages high-frequency spike camera data for refined trajectory analysis and error correction. This integrated approach compensates for systematic errors without complex physical modeling, significantly improving both interception accuracy and strategic ball striking.

### 2.2 ROBOTIC TABLE TENNIS

Robotic table tennis has long served as a benchmark task in robotics due to its requirement for real-time perception, prediction, planning, and control. Since Billingsley initiated the first robot table tennis competition in 1983 (Billingsley, 1983), the task has attracted continuous attention from the research community. Existing approaches can be broadly categorized into two groups: control-based methods and learning-based methods. **Control-based approaches** (Acosta et al., 2003; Mülling et al., 2010; Zhang et al., 2018) rely on mathematical modeling and predefined control strategies, typically following a perception-prediction-control pipeline. While these methods benefit from mathematical rigor, they often require precise calibration and struggle with adapting to environmental variations. **Learning-based approaches**, particularly reinforcement learning and imitation learning, have gained prominence recently. RL approaches (Büchler et al., 2020; Tebbe et al., 2021; Gao et al., 2020; DAmbrosio et al., 2025; Su et al., 2025) directly map sensory inputs to motor commands, offering greater adaptability. Abeyruwan et al. (Abeyruwan et al., 2023) proposed iterative sim-to-real transfer, while GoalsEye (Ding et al., 2022) employs imitation learning through demonstrations and self-supervised practice, but its sim2real reliance limits performance. Our *SpikePingpong* system differs by introducing a Fast-Slow system perception architecture and real-world imitation learning for ball striking, training directly on data collected from real-world interactions without complex human assistance or simulation dependencies, achieving superior performance and practical deployment.

## 3 METHOD

In this section, we present our robotic table tennis framework *SpikePingpong*, consisting of two integrated components as shown in Figure 2. Our approach features a Fast-Slow system for perception: System 1 employs physics-based trajectory prediction using an RGB-D camera for rapid ball detection, while System 2 leverages a high-frequency spike camera to refine predictions by compensating for real-world physical effects. For action generation, our Imitation-based Motion Planning And Control Technology (IMPACT) module generates strategic hitting motions through imitation learning, enabling tactical control over return placement.

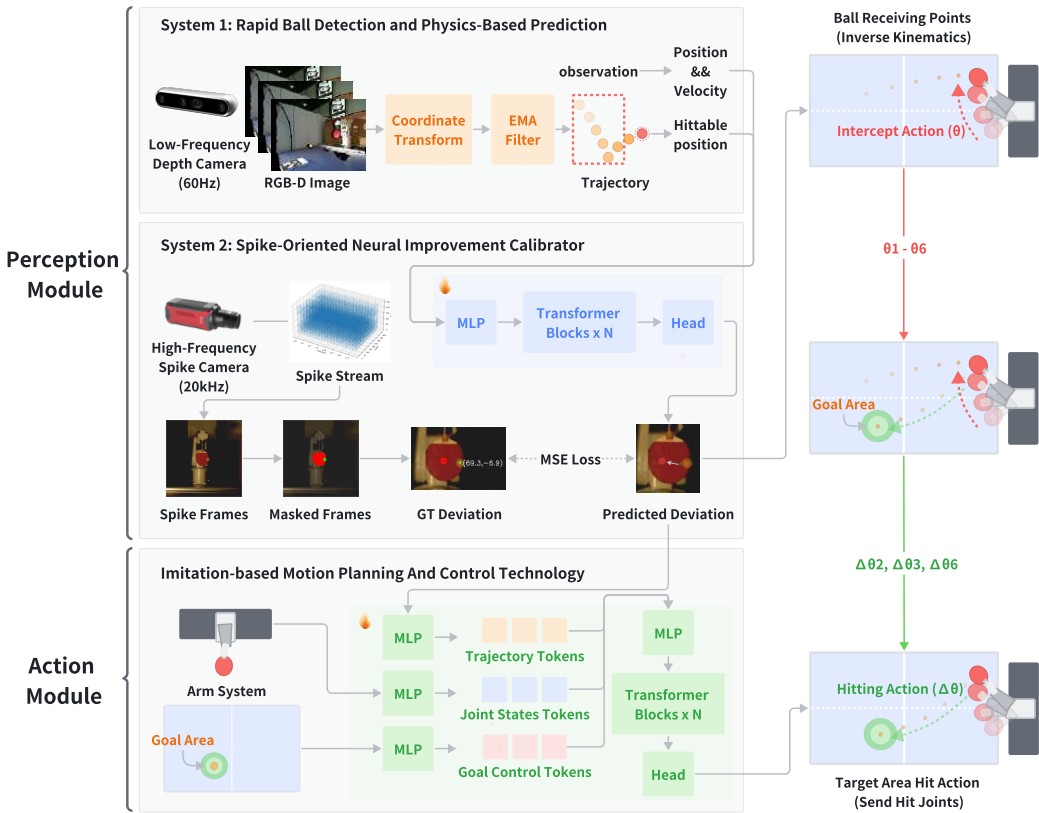

Figure 2: **Framework of *SpikePingpong*.** The system comprises two integrated components: (1) A Fast-Slow perception architecture, where System 1 delivers rapid trajectory prediction using RGB-D data, while System 2 functions as a Spike-Oriented Neural Improvement Calibrator to refine the estimated hittable position; and (2) The IMPACT module, which facilitates strategic motion planning and control, enabling tactical return placement via imitation learning.

## 3.1 FAST-SLOW SYSTEM BALL INTERCEPTION FRAMEWORK

### 3.1.1 SYSTEM 1: RAPID BALL DETECTION AND PHYSICS-BASED PREDICTION.

System 1 serves as the foundation of our Fast-Slow system architecture, providing rapid response capabilities for high-speed ball tracking through two core components: real-time ball detection and physics-based trajectory prediction. The detection module extracts ball positions from RGB-D camera streams, while the prediction module utilizes classical physics models to estimate future ball trajectories and determine optimal hittable positions. Additionally, System 1 provides essential contextual information to System 2, including ball state estimates and predicted hittable position that inform the learning-based decision-making process.

**Ball Detection.** We employ YOLOv4-tiny (bubbliiiing, 2020) for its computational efficiency, achieving detection frequencies up to 150 Hz. Our training methodology adopts a two-phase approach: initial pre-training on public datasets (Roboflow (Alexandrova et al., 2015), TT2 (desigproject, 2022), and Ping Pong Detection (pingpong, 2024)), followed by domain-specific fine-tuning. Following object detection, we perform coordinate transformation from image space to world coordinates using calibrated camera parameters, enabling accurate 3D ball position acquisition for subsequent motion planning.

**Physics-Based Trajectory Prediction.** Our physics-based approach represents another key component of System 1, enabling the system to anticipate the ball's path and determine optimal hittable positions. The prediction model employs Exponential Moving Average filtering to obtain reliable estimates of the ball's current position $(x, y, z)$ and velocity $(v_x, v_y, v_z)$. These filtered state estimates

serve as input to our physics model, which outputs the predicted hittable position $(x_{\text{hit}}, y_{\text{hit}}, z_{\text{hit}})$ and corresponding velocity $(v_x^{\text{hit}}, v_y^{\text{hit}}, v_z^{\text{hit}})$. We calculate the time $t$ required for the ball to reach the pre-determined hitting plane at $y_{\text{hit}}$: $t = \frac{y_{\text{hit}} - y}{v_y}$. Using this time value, we predict the x-coordinate at the hittable position: $x_{\text{hit}} = x + v_x \cdot t$. If $x_{\text{hit}}$ falls outside the robot's operational workspace, the ball is classified as unhittable. For the z-coordinate prediction, we consider two scenarios:

- **Direct trajectory:** If the ball doesn't contact the table before reaching $y_{\text{hit}}$, we compute $z_{\text{hit}} = z + v_z \cdot t + \frac{1}{2}gt^2$. When $z_{\text{hit}} > h_{table}$, no rebound occurs.
- **Rebound trajectory:** If the ball impacts the table, we calculate the rebound time $t_{\text{rb}}$ by solving: $z + v_z \cdot t_{\text{rb}} + \frac{1}{2}gt_{\text{rb}}^2 = h_{table}$.
  We then determine the impact velocity $v_{z,\text{in}}$ and post-rebound velocity $v_{z,\text{out}}$ using:

$$v_{z,\text{in}} = -\sqrt{-2g\left(z - h_{table}\right) + v_z^2}, \tag{1}$$
$$v_{z,\text{out}} = -e \cdot v_{z,\text{in}}. \tag{2}$$

where $e$ represents the coefficient of restitution. The system further evaluates potential secondary rebounds to determine the final $z_{\text{hit}}$ and $v_z^{\text{hit}}$.

### 3.1.2 SYSTEM 2: SPIKE-ORIENTED NEURAL IMPROVEMENT CALIBRATOR

System 2 serves as the learning-based enhancement layer of our Fast-Slow system architecture, addressing the limitations of physics-based predictions through neural calibration. While System 1 provides rapid trajectory estimation under ideal conditions, real-world scenarios introduce deviations due to air resistance, ball spin, and sensor noise that simplified physics models cannot capture. To bridge this gap, System 2 functions as a Spike-Oriented Neural Improvement Calibrator, which learns to predict the systematic discrepancy between System 1's theoretical hittable position and actual optimal interception position. By leveraging high-frequency spike camera observations and contextual information from System 1, System 2 provides precise calibration corrections that significantly enhance overall system accuracy.

**Data Collection.** System 2 integrates ball trajectory data, velocity measurements, and physics-based predictions to precisely quantify the discrepancy between theoretical and actual hittable positions. For training purposes, we developed an extensive dataset that meticulously documents the systematic variations between physics-model predictions and empirically observed real-world interception positions, enabling our system to learn these complex error patterns.

Specifically, during each trial, we systematically record the ball's 3D position and velocity vectors throughout its trajectory, along with the corresponding hittable position predicted by System 1's physics-based model. Based on these predictions, we compute the required joint angles through inverse kinematics and execute the corresponding robotic arm motion to position the paddle center at the theoretically optimal hittable position. Subsequently, a Spike camera (Dong et al., 2021) captures images of the actual ball-paddle interaction at the moment of contact. The pixel distance between the ball's observed position and the paddle center in these images provides a quantitative measure of the spatial deviation, which serves as the ground truth for System 2.

**Network Architecture.** System 2's network processes three input modalities: historical position vectors $\mathbf{p}_i \in \mathbb{R}^{K \times 3}$, velocity vectors $\mathbf{v}_i \in \mathbb{R}^{K \times 3}$ from the preceding $K$ frames, and the physics-based predicted hittable position $\mathbf{h}_i \in \mathbb{R}^3$. Each modality is processed through dedicated MLPs with ReLU activation and dropout regularization for feature extraction. The concatenated features are then processed through a Transformer (Vaswani et al., 2017) encoder to capture temporal dependencies and contextual relationships across trajectory segments. Finally, a regression head with fully connected layers maps the refined representation to the predicted deviation vector.

**Training Objective.** We optimize the model using the mean squared error (MSE) loss function, which minimizes the difference between the predicted and ground-truth deviation vectors:

$$L_{MSE}(\theta) = \frac{1}{N}\sum_{i=1}^{N}||\hat{D}_i - D_i||^2; \text{where } \hat{D}_i = f_\theta([p_i, v_i, h_i]). \tag{3}$$

$f_\theta$ represents the neural network with parameters $\theta$, $p_i \in \mathbb{R}^{K \times 3}$ denotes the position history, $v_i \in \mathbb{R}^{K \times 3}$ denotes the velocity history, $h_i \in \mathbb{R}^3$ denotes the expected hittable position, $D_i \in \mathbb{R}^2$ is the ground truth deviation vector, and $\hat{D}_i \in \mathbb{R}^2$ is the predicted deviation vector.

Drawing inspiration from dual-system theory in cognitive science (Kahneman, 2011), this Fast-Slow system architecture combines the speed of immediate heuristic reasoning with the accuracy of experience-based learning, ensuring both real-time operation and precise task execution. Once trained, System 2 operates as a lightweight neural predictor that directly estimates deviation vectors from trajectory features without requiring spike camera feedback during deployment. This design enables the system to benefit from high-fidelity spike-based training data while maintaining computational efficiency and real-time performance in operational scenarios.

## 3.2 IMPACT: IMITATION-BASED MOTION PLANNING AND CONTROL TECHNOLOGY

Building upon our Fast-Slow system framework, where System 1 provides rapid physics-based predictions and System 2 delivers precise neural calibration through spike-oriented improvement, we introduce IMPACT (Imitation-based Motion Planning And Control Technology) for strategic striking behaviors. This module learns tactical ball-striking through imitation learning, enabling strategic returns that achieve targeted gameplay beyond mere interception.

**Data Collection.**   The IMPACT module requires high-quality training data that captures the complete striking process from ball trajectory to final landing outcomes. Firstly, we record the incoming ball trajectory and estimate the optimal hitting position using our Fast-Slow system framework. Based on this prediction, we compute the corresponding robot joint configurations through inverse kinematics and position the robotic arm accordingly. To generate diverse striking behaviors, we apply random angular perturbations to three critical robot joints before executing the stroke. We retain only successful trials where the ball returns to the opponent's side, recording the perturbed joint angles and resulting landing positions. Each sample is labeled according to its specific landing region for fine-grained strategic control. This approach is highly efficient compared to teleoperation methods (Takada et al., 2022), as it leverages accurate hitting predictions to automatically position the robot, significantly reducing collection time while ensuring consistent data quality.

**Network Architecture.**   The IMPACT module employs a transformer-based neural network that processes three input modalities: ball trajectory sequences, robot joint configurations, and desired landing region specifications (see Figure 2). Each modality is independently encoded into token representations using dedicated multi-layer perceptrons (MLPs). These tokens are concatenated to form a unified input sequence, which is processed by a Transformer encoder that leverages self-attention mechanisms to capture inter-modal dependencies. The network outputs optimal joint angle adjustments for precise ball placement control, effectively integrating trajectory information, kinematic constraints, and strategic objectives within a unified framework.

**Training Objective.**   To train the model, we employ the mean squared error (MSE) loss function, which minimizes the discrepancy between the predicted and ground-truth joint adjustments:

$$L_{MSE}(\theta') = \frac{1}{N} \sum_{i=1}^{N} ||\hat{J}_i - J_i||^2; \text{ where } \hat{J}_i = f_{\theta'}([p_i, v_i, j_i, c_i]). \tag{4}$$

Here, $f_{\theta'}$ represents the neural network with parameters $\theta'$, $p_i \in \mathbb{R}^{K \times 3}$ and $v_i \in \mathbb{R}^{K \times 3}$ denote the ball's position and velocity history, $j_i \in \mathbb{R}^6$ represents the 6-DOF robot joint configuration, $c_i \in \mathbb{R}^4$ represents the one-hot encoded control signal for the desired landing region, $J_i \in \mathbb{R}^3$ is the ground truth adjustment vector, and $\hat{J}_i \in \mathbb{R}^3$ is the predicted joint adjustment vector.

Through this imitation learning framework, IMPACT enables the robot to dynamically adapt its striking strategy to varying ball trajectories while precisely targeting specific regions on the opponent's court. This capability transforms the system from merely achieving technical accuracy to executing sophisticated tactical gameplay, significantly enhancing the robot's strategic competitiveness in table tennis matches.

## 4 EXPERIMENTS

We present comprehensive evaluations of *SpikePingpong* across multiple dimensions of table tennis performance. We begin with our experimental setup (Section 4.1), followed by the main results evaluating contact precision, tactical spatial control in both single and sequential tasks, and robustness under out-of-distribution and human-interaction scenarios (Section 4.2). Finally, we conduct ablation studies to validate our architectural choices (Section 4.3).

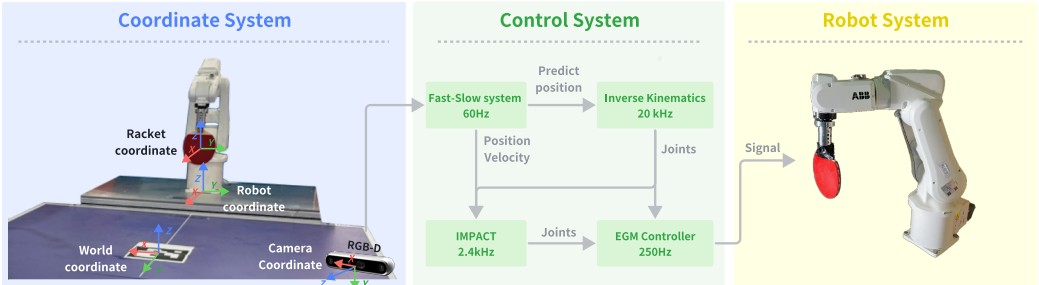

Figure 3: **System Overview.** Our system integrates three key subsystems: (1) a coordinate system for spatial tracking and transformation, (2) a multi-frequency control system with Fast-Slow system, IMPACT, and an EGM controller, and (3) a robot system based on the ABB IRB-120 arm equipped with a standard table tennis racket.

### 4.1 EXPERIMENT SETTING

**Dataset.** We introduce the training Dataset, collected using an automated launcher with highly randomized ball dynamics (spin, speed, and placement). It comprises two subsets: (1) 1k samples of trajectory and ground-truth contact deviation data, synchronously captured via RGB-D (60 Hz) and spike cameras (20 kHz); and (2) 2k expert return demonstrations, mapping ball trajectories to robot joint configurations for targeted placement. Further details are in Appendix D.

**Implementation Details.** As shown in Fig. 3, our system consists of three integrated subsystems. The coordinate system handles spatial transformations between world, camera, robot, and racket coordinates using ArUco marker calibration. The control system operates at multiple frequencies: Fast-Slow system processes trajectory data at 60Hz and converts predictions to joint configurations via inverse kinematics at 20kHz, while IMPACT operates at 2.4kHz for striking adjustments with commands transmitted to the EGM controller at 250Hz. The robot system employs an ABB IRB-120 robotic arm with a standard table tennis racket, implemented on a workstation with NVIDIA RTX 4090 GPU. Training details are provided in the Appendix E.

**Baselines.** We benchmark against ACT (Zhao et al., 2023) and Diffusion Policy (Chi et al., 2023). To ensure fair comparison, we optimized both baselines by using state-based inputs (identical to IMPACT) instead of raw images to eliminate visual latency. Additionally, we accelerated Diffusion Policy using a 10-step DDIM sampler. All methods employ a synchronous just-in-time one-shot inference strategy, executing a single forward pass at the latest decision point to leverage the most accurate state estimation.

**Evaluation.** We evaluate system performance through offline validation, online physical testing, and out-of-distribution (OOD) scenarios. Contact precision (MAE/RMSE) is measured on a held-out test set (80/10/10 split). Landing accuracy is assessed across four table regions (A, B, C, D) using two thresholds: 30cm (primary) and 20cm (high precision). Online evaluations utilize a ball launcher generating randomized trajectories (spin, speed, and placement) to ensure distinctness from the training set. To test generalization, we conduct OOD experiments with relocated launcher positions and zero-shot transfer against unseen human opponents. These standardized metrics ensure a rigorous assessment in the absence of publicly available hardware baselines.

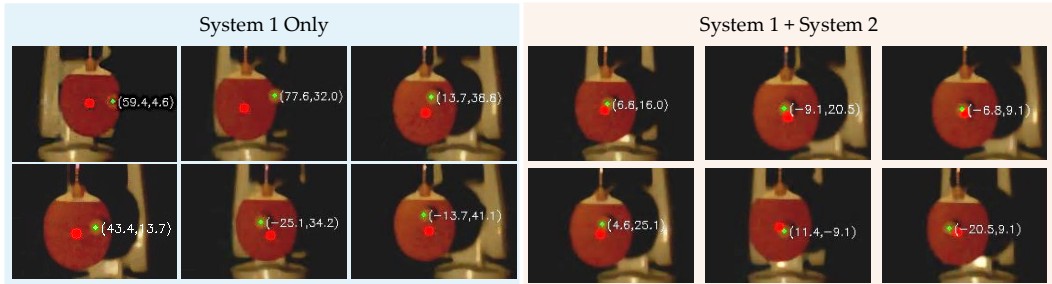

Figure 4: **Visualization of ball-racket contact precision with and without System 2.** Spike-based camera images show the ball center (green) and racket center (red) at contact moment. The reduced offset with the Fast-Slow System demonstrates improved ball interception accuracy.

Table 1: **Ball Hittable Position Prediction Error.** Our Fast-Slow system approach achieves superior precision in predicting the actual ball-racket contact point across both axes.

| Method | Y-axis | | Z-axis | | **Overall** | |
|---|---|---|---|---|---|---|
| | MAE | RMSE | MAE | RMSE | **MAE** | **RMSE** |
| System 1 Only | 53.65 | 60.39 | 34.62 | 38.45 | 44.13 | 50.62 |
| RNN-based Method (Elman, 1990) | 24.10 | 24.89 | 21.50 | 22.52 | 22.80 | 23.73 |
| **System 1 + System 2 (Ours)** | **9.87** | **11.16** | **14.82** | **16.10** | **12.34** | **13.85** |

## 4.2 MAIN RESULTS

**System Precision and Efficiency** Accurate and timely ball interception is the prerequisite for any successful table tennis system. By leveraging our Fast-Slow system design, *SpikePingpong* achieves superior contact precision, with an MAE of 12.34 and an RMSE of 13.85 for deviation prediction (Table 1). As visualized in Figure 4, the spike camera captures reveal minimal spatial separation between the ball and racket centers at the moment of contact.

Crucially, to translate this spatial accuracy into physical actuation, our system must operate with minimal latency. As shown in Table 2, *SpikePingpong* achieves an inference time of just 0.407ms, dramatically outpacing Diffusion Policy and ACT. This millisecond-level responsiveness ensures sufficient time for the robotic arm's physical execution.

Table 2: **Computational Performance Comparison.** Average inference times in milliseconds for generating return actions across different methods.

| Method | Inference time (ms) |
|---|---|
| Diffusion Policy (Chi et al., 2023) | 25.18 |
| ACT (Zhao et al., 2023) | 7.15 |
| **SpikePingpong** | **0.407** |

**Tactical Spatial Control** We conducted rigorous comparative evaluations against established baseline methods across four strategically positioned target regions (A, B, C, and D). Table 3 presents success rates across various target regions and precision thresholds. *SpikePingpong* exhibits exceptional performance, achieving an average success rate of 92% within the 30cm accuracy zone and 70% within the 20cm high-precision zone, with consistent performance across all target regions. Our system substantially outperforms the human average (53% at 30cm) and all baseline methods. Notably, while baselines like ACT (Zhao et al., 2023) performed better without visual inputs (from 12% to 19%), our approach still demonstrates a significant performance leap.

Beyond isolated returns, to evaluate our system's ability to execute tactical sequences, we designed experiments involving random target sequences spanning 100 consecutive returns across four target regions, with 25 targets per region presented in random order. We define success using a 30cm precision threshold, with additional 20cm precision metrics provided for detailed analysis. Table 4 presents the results of our sequential target execution experiment. *SpikePingpong* achieves an overall sequence success rate of 78%, significantly outperforming the human baseline of 45%. Our system demonstrates balanced performance across all court regions with consistent precision under stricter 20cm thresholds. These results demonstrate that our system maintains high precision while executing complex, extended tactical sequences that surpass human-level performance, representing a significant advancement toward sustained strategic gameplay rather than merely returning balls.

Table 3: **Single-Target Return Accuracy (%).** Success rates for ball striking across four distinct target regions (A-D) at both 30cm and 20cm precision thresholds. The table compares human players, previous robotic approaches, and our SpikePingpong system. Higher percentages indicate better performance. Standard deviations are denoted in subscripts.

| Method | Region A | | Region B | | Region C | | Region D | | Average | |
|---|---|---|---|---|---|---|---|---|---|---|
| | 30cm | 20cm | 30cm | 20cm | 30cm | 20cm | 30cm | 20cm | 30cm | 20cm |
| Human Avg. | $48_{\pm6}$ | $28_{\pm4}$ | $52_{\pm5}$ | $32_{\pm3}$ | $56_{\pm7}$ | $38_{\pm5}$ | $54_{\pm4}$ | $34_{\pm6}$ | $53_{\pm6}$ | $33_{\pm5}$ |
| Diffusion Policy (w/ vision) | $3_{\pm2}$ | $1_{\pm1}$ | $4_{\pm3}$ | $0_{\pm0}$ | $2_{\pm2}$ | $1_{\pm1}$ | $3_{\pm2}$ | $0_{\pm0}$ | $3_{\pm2}$ | $1_{\pm1}$ |
| Diffusion Policy (w/o vision) | $6_{\pm3}$ | $2_{\pm1}$ | $7_{\pm4}$ | $2_{\pm2}$ | $5_{\pm2}$ | $1_{\pm1}$ | $6_{\pm3}$ | $1_{\pm1}$ | $6_{\pm3}$ | $2_{\pm1}$ |
| ACT (w/ vision) | $11_{\pm4}$ | $4_{\pm2}$ | $12_{\pm5}$ | $4_{\pm1}$ | $10_{\pm3}$ | $2_{\pm1}$ | $14_{\pm4}$ | $5_{\pm2}$ | $12_{\pm4}$ | $4_{\pm2}$ |
| ACT (w/o vision) | $18_{\pm5}$ | $7_{\pm3}$ | $20_{\pm6}$ | $8_{\pm3}$ | $17_{\pm4}$ | $6_{\pm2}$ | $19_{\pm5}$ | $7_{\pm2}$ | $19_{\pm5}$ | $7_{\pm3}$ |
| **SpikePingpong** | $\mathbf{91}_{\pm3}$ | $\mathbf{69}_{\pm4}$ | $\mathbf{93}_{\pm2}$ | $\mathbf{72}_{\pm3}$ | $\mathbf{92}_{\pm4}$ | $\mathbf{70}_{\pm5}$ | $\mathbf{93}_{\pm3}$ | $\mathbf{71}_{\pm4}$ | $\mathbf{92}_{\pm3}$ | $\mathbf{70}_{\pm4}$ |

Table 4: **Sequential Target Execution Performance (%).** Success rates for 100-shot random target sequences. The overall rate column shows the percentage of individual shots successfully reaching their designated targets (30cm), while preceding columns present region-specific success rates. Standard deviations for the overall rate are denoted in subscripts.

| Method | Region A | | Region B | | Region C | | Region D | | Success Rate |
|---|---|---|---|---|---|---|---|---|---|
| | 30cm | 20cm | 30cm | 20cm | 30cm | 20cm | 30cm | 20cm | |
| Human Avg. | 44 | 26 | 47 | 28 | 43 | 25 | 46 | 29 | $45_{\pm5}$ |
| Diffusion Policy (w/ vision) | 1 | 0 | 2 | 0 | 1 | 0 | 1 | 0 | $1_{\pm1}$ |
| Diffusion Policy (w/o vision) | 2 | 1 | 3 | 1 | 2 | 0 | 2 | 0 | $2_{\pm2}$ |
| ACT (w/ vision) | 7 | 2 | 9 | 3 | 6 | 2 | 10 | 3 | $8_{\pm3}$ |
| ACT (w/o vision) | 14 | 5 | 16 | 6 | 13 | 4 | 17 | 5 | $15_{\pm4}$ |
| **SpikePingpong** | **76** | **52** | **79** | **54** | **77** | **51** | **80** | **55** | $\mathbf{78}_{\pm3}$ |

**Robustness and Out-of-Distribution Generalization**    To evaluate the generalization capability of our policy beyond the training distribution, we conducted a challenging out-of-distribution (OOD) experiment. While all training data was collected with the ball launcher at a fixed central position on the opponent's table, for the OOD test, we physically moved the launcher to two new, unseen off-center positions. This fundamentally altered the entire distribution of incoming ball trajectories, including their angles, speeds, and bounce locations.

As shown in Table 5, while there is a predictable drop in performance compared to the in-distribution setting, our system maintained a remarkable average success rate of 74% for 30cm targets. This result is significant, as the off-center launcher positions create entirely novel trajectories that the policy has never encountered. The sustained high performance strongly suggests that our policy has learned a robust, underlying model of ball dynamics and striking control, rather than simply memorizing patterns from the training data. This validates the generalization capabilities of our framework.

Table 5: **Out-of-Distribution Generalization.** Success rates on seen vs. unseen trajectory distributions.

| Condition | Avg. Success Rate | |
|---|---|---|
| | 30cm (%) | 20cm (%) |
| In-Distribution | $92_{\pm3}$ | $70_{\pm4}$ |
| Out-of-Dist. | $74_{\pm5}$ | $52_{\pm6}$ |

Furthermore, to push the boundaries of generalization, we tested our system's ability to adapt to and generalize from complex, human-generated trajectories. Unlike the structured patterns from a robotic launcher, human shots introduce significant, unstructured variability, representing a more challenging data distribution. We conducted a two-stage experiment to assess this capability, with results summarized in Table 6. First, to test adaptability, we fine-tuned our model on a small dataset of 100 demonstrations from a single human player (Person A). As shown in the "Seen Player" rows of Table 6, when tested on this same player, the system achieved a meaningful success rate of 47% for 30cm targets, demonstrating strong sample efficiency. Next, to perform a stricter test of generalization, we evaluated this fine-tuned model in a zero-shot setting against a new, unseen player (Person B). The results in the "Unseen Player" rows show a drop in performance to 31%, as expected in a challenging OOD scenario. Nevertheless, achieving over 31% accuracy on a completely new person without any specific fine-tuning is a non-trivial result. It provides promising evidence that

Table 6: **Adaptation and Generalization to Human Demonstrations (%).** Success rates on precision targeting tasks. "Seen Player" refers to testing on the same human demonstrator whose data was used for fine-tuning. "Unseen Player" refers to zero-shot testing on a new human player. Standard deviations are denoted in subscripts.

| Precision | Condition | Region A | Region B | Region C | Region D | Average |
|---|---|---|---|---|---|---|
| 30cm | Seen Player (Person A) | $51_{\pm 5}$ | $47_{\pm 3}$ | $42_{\pm 3}$ | $50_{\pm 4}$ | $\mathbf{47_{\pm 3}}$ |
| | Unseen Player (Person B) | $35_{\pm 6}$ | $31_{\pm 5}$ | $27_{\pm 5}$ | $32_{\pm 4}$ | $\mathbf{31_{\pm 5}}$ |
| 20cm | Seen Player (Person A) | $29_{\pm 2}$ | $23_{\pm 4}$ | $27_{\pm 2}$ | $28_{\pm 3}$ | $\mathbf{27_{\pm 3}}$ |
| | Unseen Player (Person B) | $18_{\pm 3}$ | $13_{\pm 4}$ | $16_{\pm 3}$ | $14_{\pm 4}$ | $\mathbf{15_{\pm 4}}$ |

Table 7: **Ablation Study (%).** Performance comparison of different trajectory prediction components in our SpikePingpong system across two distinct tasks.

| Method | Region A | Region B | Region C | Region D | Overall |
|---|---|---|---|---|---|
| **Task 1: Single-Target Return Accuracy** | | | | | |
| System 1 + IMPACT | $22_{\pm 6}$ | $25_{\pm 5}$ | $21_{\pm 7}$ | $24_{\pm 5}$ | $23_{\pm 6}$ |
| RNN (Elman, 1990) + IMPACT | $65_{\pm 5}$ | $68_{\pm 4}$ | $66_{\pm 6}$ | $69_{\pm 3}$ | $67_{\pm 5}$ |
| **SpikePingpong** | $\mathbf{91_{\pm 3}}$ | $\mathbf{93_{\pm 2}}$ | $\mathbf{92_{\pm 4}}$ | $\mathbf{93_{\pm 3}}$ | $\mathbf{92_{\pm 3}}$ |
| **Task 2: Sequential Target Execution Performance** | | | | | |
| System 1 + IMPACT | $18_{\pm 5}$ | $20_{\pm 4}$ | $17_{\pm 6}$ | $19_{\pm 5}$ | $15_{\pm 4}$ |
| RNN (Elman, 1990) + IMPACT | $55_{\pm 4}$ | $58_{\pm 5}$ | $54_{\pm 6}$ | $57_{\pm 4}$ | $52_{\pm 6}$ |
| **SpikePingpong** | $\mathbf{76_{\pm 4}}$ | $\mathbf{79_{\pm 3}}$ | $\mathbf{77_{\pm 4}}$ | $\mathbf{80_{\pm 3}}$ | $\mathbf{78_{\pm 3}}$ |

our model captures generalizable features of human-like dynamics rather than simply overfitting to an individual's style, highlighting its potential for collaborative human-robot scenarios.

## 4.3 ABLATION STUDY

We conducted ablation experiments across four target regions (A, B, C, D) within a 30cm radius threshold to evaluate each component's contribution. Our Fast-Slow system architecture achieves 92% accuracy in single-target returns, representing a 25-percentage-point improvement over the RNN (Elman, 1990) baseline. For sequential target execution, our Fast-Slow system achieves 78% success rate compared to 52% for the RNN-based method (Elman, 1990). This substantial enhancement stems from the system's ability to account for ball-racket interaction sensitivity, where identical striking motions can produce vastly different trajectories depending on precise contact positioning. The consistent improvements across both single-target and sequential tasks validate the robustness of our architectural design.

## 5 CONCLUSIONS

In this paper, we presented *SpikePingpong*, a high-precision robotic table tennis system integrating spike vision and advanced control techniques from a cognitive perspective. Our Fast-Slow system architecture emulates human visual perception processes for enhanced ball detection and trajectory prediction, while IMPACT handles strategic motion planning for accurate ball striking to specified target regions. Experiments demonstrate superior performance with 92% success in the primary accuracy zone and 70% in the high-precision zone, significantly outperforming previous systems. *SpikePingpong* enables real-time decision-making while executing extended tactical sequences with a 78% success rate, showcasing sustained strategic gameplay capabilities. The success of our approach stems from the Fast-Slow architecture that combines rapid physics-based prediction with neural calibration, and the IMPACT module that enables strategic ball placement through imitation learning. Beyond table tennis, the core competencies developed in this work, including high-speed object tracking, precision manipulation, and adaptive control, demonstrate broad applicability to industrial automation, medical robotics, and aerospace systems. This work advances robotic capabilities in time-critical manipulation tasks requiring precise spatiotemporal coordination.

ETHICS STATEMENT

We acknowledge and adhere to the ICLR Code of Ethics. All data collection focused solely on ball trajectories and paddle movements without storing personally identifiable information. This research contributes to recreational robotics applications and presents minimal ethical concerns when developed responsibly.

REPRODUCIBILITY STATEMENT

To ensure the reproducibility of our work, we provide comprehensive implementation details and experimental specifications throughout the paper and supplementary materials. The experimental setup, including hardware specifications, data collection protocols, and evaluation metrics, is thoroughly documented in Appendix B. The dataset collection methodology and preprocessing steps are detailed in Appendix D, enabling replication of our data generation process. Appendix E contains detailed network architectures, hyperparameters, and training procedures for all components of our system. We've provided the code for our method in the supplementary material. Our codebase, including model implementations, training scripts, and evaluation protocols, will be made publicly available upon publication.

ACKNOWLEDGEMENT

This work was supported by the National Natural Science Foundation of China (62476011), and by the Beijing Natural Science Foundation (L252060).

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

# A    OVERVIEW

Due to space limitations, we provide comprehensive implementation details and additional experimental validation in the appendix. Section B presents our complete system architecture, including the ball launcher, multi-camera vision setup, and robotic execution components. Section C details our real-time perception pipeline with YOLOv4-tiny ball detection, physics-based filtering, and trajectory prediction. Section D describes our two specialized datasets for the Fast-Slow System and IMPACT models. Section E provides training specifications. Section F presents extended validation, including ultra-high precision evaluation and human demonstration integration. Section G provides a comprehensive failure case analysis to identify system limitations and improvement opportunities. Section H discusses current system limitations and outlines future research directions for enhanced spin modeling, human player adaptability, and strategic gameplay planning. Section I provides a transparent disclosure of the limited use of Large Language Models in manuscript preparation. Additionally, we provide supplementary videos demonstrating the robotic system's real-time ball-hitting performance and trajectory interception capabilities.

# B    SYSTEM OVERVIEW

Our system consists of a ball launcher, high-speed camera, depth camera, RGB camera, and robotic arm, designed to achieve an automated table tennis playing system.

## B.1    BALL LAUNCHING SYSTEM

We employ the intelligent table tennis robot PONGBOT NOVA as our ball launching system. This table-mounted launcher can generate topspin and backspin, with precise landing point control ranging from -2 to +2, and adjustable ball speed between levels 1-3, providing stable and controllable ball trajectories for our experiments.

## B.2    VISION SYSTEM

The vision system comprises three cameras, each dedicated to different tasks:

- **High-Frequency Camera**: Spike M1K40-H2-Gen3 with a resolution of 1000×1000 and a maximum frame rate of 20,000 fps. The device is specifically configured to capture the instantaneous contact between the ball and the paddle, ensuring high temporal resolution image data for detailed stroke analysis.

- **Depth Camera**: Intel RealSense D455 with a resolution of 640×480 and a frame rate of 60Hz. This camera is calibrated using ArUco markers and employs YOLO (bubbliiiing, 2020) model for ball detection. The system transforms detected pixel coordinates into world coordinates through intrinsic and extrinsic parameters, enabling real-time tracking of the ball's position.

- **RGB Camera**: Intel RealSense LiDAR Camera L515 with a resolution of 960×540 and a frame rate of 60Hz. It primarily detects the landing position of the ball after being hit by the robot, providing feedback for evaluating stroke effectiveness.

Figure 5 illustrates the visual comparison between spike camera and conventional RGB camera captures. Due to its ultra-high frame rate, the spike camera eliminates motion blur entirely, providing crisp imagery of fast-moving objects that would appear blurred in standard cameras, which is crucial for precise ball trajectory analysis during high-speed table tennis gameplay.

## B.3    MECHANICAL EXECUTION SYSTEM

The execution system utilizes an ABB IRB 120 6 DoF (six-degree-of-freedom) robotic arm with maximum joint rotation speeds of 250°/s, 250°/s, 250°/s, 320°/s, 320°/s, and 420°/s, respectively. The arm is equipped with a standard table tennis paddle at its end-effector to execute the optimal hitting motion calculated by the system.

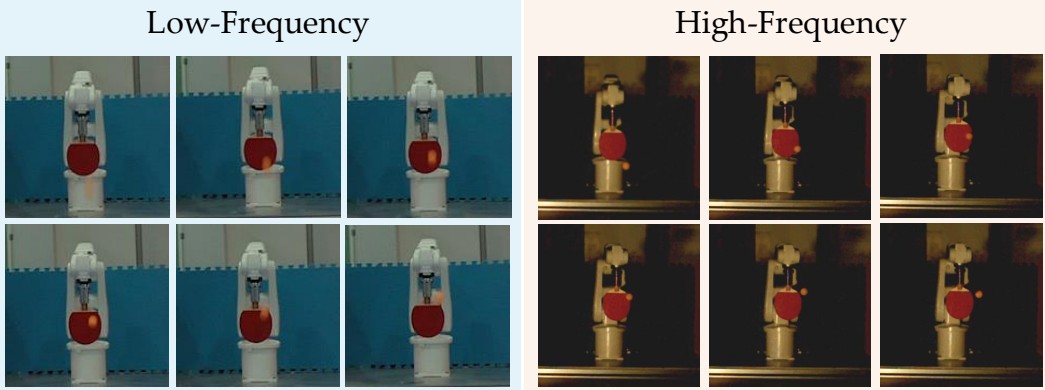

Figure 5: **Comparison of ball capture qualit.** Conventional RGB camera with motion blur at 60 fps, Spike camera with crisp imagery at 20,000 fps, demonstrating the advantage of ultra-high frame rate for fast-moving object detection.

This system achieves real-time tracking and precise hitting of table tennis balls through the tight integration of visual perception, trajectory prediction, and motion planning, providing a comprehensive experimental platform for table tennis robotics research.

## C   DETAILS OF BALL DETECTION AND TRAJECTORY PREDICTION

### C.1   BALL DETECTION

In the context of high-speed robotic table tennis, where accurate timing and spatial awareness are critical, a high-frequency vision system is required to continuously track the ball's position for real-time trajectory estimation and manipulation control. Thus, we chose YOLOv4-tiny (bubbliiiing, 2020) due to its lightweight design and computational efficiency, enabling a detection frequency of up to **150 Hz**, which is crucial for the precise and timely interaction with the fast-moving ball. The initial phase of our research involves the supervised training of the detection model using publicly available datasets: Roboflow (Alexandrova et al., 2015), TT2 (desigproject, 2022), and Ping Pong Detection (pingpong, 2024). A two-phase training strategy is implemented to enhance model robustness. In the first phase, the model is trained on the complete dataset. In the second phase, samples that were misdetected during the initial training are selectively sampled and assigned higher weights for fine-tuning, with the goal of improving the detector's accuracy on difficult instances. Our empirical validation is conducted using an Intel RealSense D455 RGB-D camera, where we employ the optimized lightweight detection model. This setup achieves a detection accuracy that exceeds 99.8%, facilitating rapid and precise 2D positional detection. The results are shown in Figure 6.

### C.2   MOTION STATE ESTIMATION FILTER

To accurately track the ball's motion state, we designed a filtering algorithm that combines physical models with measurement data. This algorithm not only smooths noise in the ball position data but also provides accurate velocity estimates, establishing a foundation for subsequent trajectory prediction and interception planning.

**Exponential Moving Average Filter**   We implemented a physics-based Exponential Moving Average (EMA) filter that simultaneously estimates both position and velocity by combining system dynamics models with real-time observation data. The core formula of the filter is:

$$\hat{x}_t = (1 - \alpha) \cdot f(\hat{x}_{t-1}) + \alpha \cdot z_t \tag{5}$$

where $\hat{x}_t$ is the current state estimate, $f(\hat{x}_{t-1})$ is the dynamics prediction based on the previous state estimate, $z_t$ is the current observation, $\alpha$ is the mixing constant that determines the weight ratio between observation and prediction.

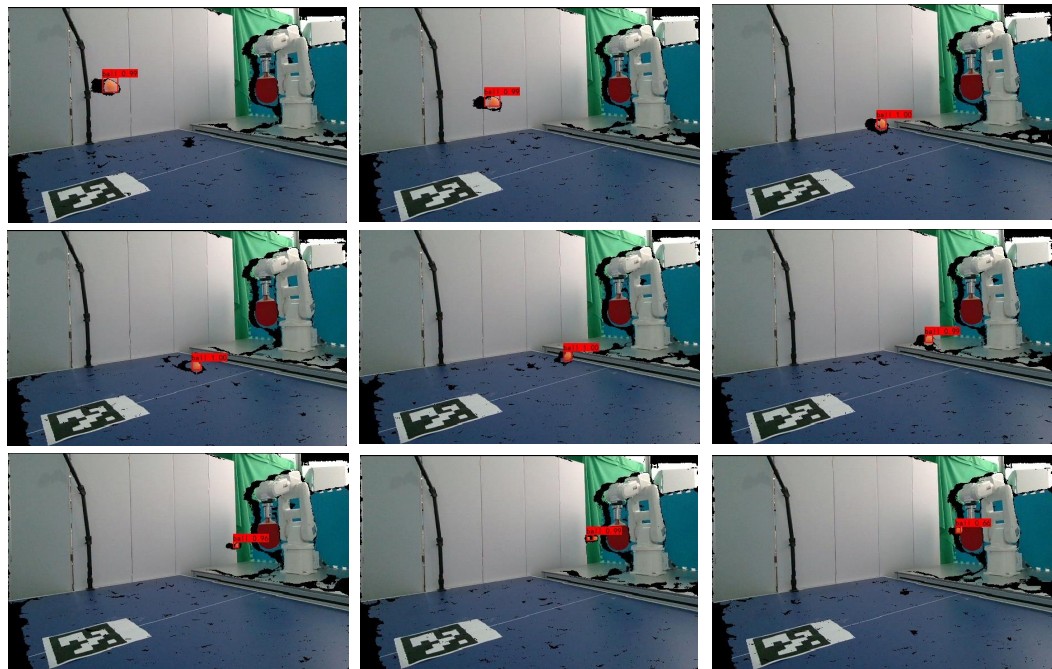

Figure 6: **Visualization of ball detection.**The sequence shows the ball's trajectory at different time points with accurately placed bounding boxes around the detected ball. Our YOLOv4-tiny model consistently identifies the ball's position even during high-speed motion, demonstrating the robustness of our detection approach under various lighting conditions and ball velocities.

For the ball's free-fall motion, we adopted standard ballistic equations as the dynamics model. Specifically, the prediction equations for position and velocity are:

$$\hat{p}_t = \hat{p}_{t-1} + \hat{v}_{t-1} \cdot \Delta t + \frac{1}{2} a \cdot \Delta t^2, \tag{6}$$

$$\hat{v}_t = \hat{v}_{t-1} + a \cdot \Delta t. \tag{7}$$

where, $\hat{p}_t$ is the position estimate, $\hat{v}_t$ is the velocity estimate, $\Delta t$ is the time step, $a$ is the acceleration (gravity acceleration -9.81 m/s² in the z-direction, 0 in x and y directions).

Considering the different motion characteristics of the ball in different directions, we applied different mixing constants for the x, y, and z directions: $\alpha_x = 0.15, \alpha_y = 0.15, \alpha_z = 0.25$. The larger mixing constant for the z-direction was chosen to better adapt to the faster velocity changes in the vertical direction due to gravity. Additionally, we implemented special handling for the ascending phase in the z-direction to more accurately capture the motion characteristics of the initial segment of the parabolic trajectory.

**Implementation Details**   The system maintains a fixed-length (10 frames) history data queue for calculating initial velocity and handling data interruptions. When a data stream interruption exceeding 1 second is detected, the filter automatically resets the historical data to avoid the influence of outdated information on current estimates. This adaptive mechanism enables the system to quickly resume normal operation when the ball reappears or a new throw begins.

**Performance Visualization**   Experiments show that this filter effectively smooths noise in the raw position data while providing accurate velocity estimates. Since the ball detection algorithm already provides relatively accurate position information, the filter primarily serves to refine and stabilize estimates, particularly excelling in velocity calculation. The filtered trajectory data exhibits smooth parabolic characteristics consistent with physical laws, providing a reliable foundation for subsequent trajectory prediction as shown in Figure 7.

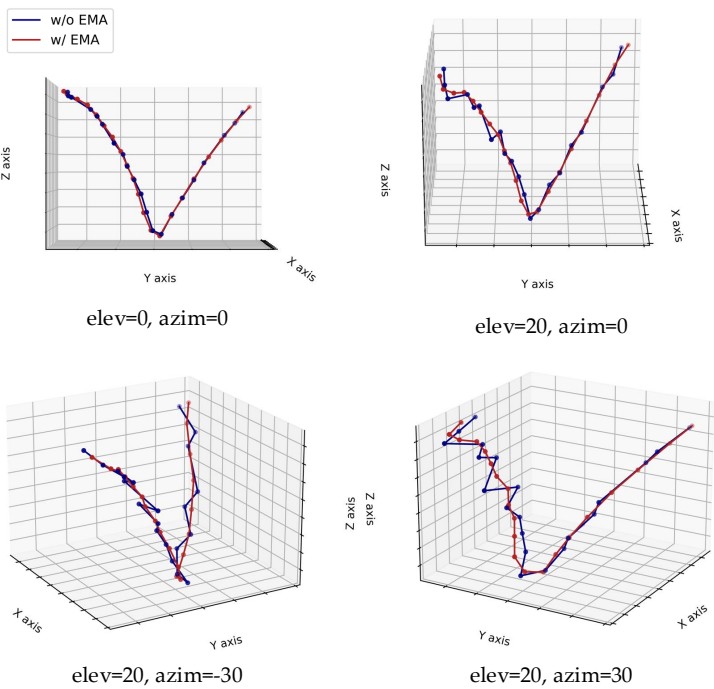

Figure 7: **Comparison of ball trajectory with and without EMA filtering.** The figure displays trajectories from multiple viewing angles, with red curves showing raw detection data and blue curves showing EMA-filtered results. The filtering effectively removes noise while maintaining the ball's natural parabolic motion, providing more reliable data for trajectory prediction.

### C.3 HITTABLE POSITION PREDICTION

To enable the robot to intercept the ball effectively, we developed a trajectory prediction algorithm that estimates where and when the ball will reach a hittable position. This algorithm leverages the filtered position and velocity data from our EMA filter to project the ball's future path.

**Ballistic Model Implementation**    Our prediction algorithm employs a simplified ballistic model that accounts for gravitational acceleration (9.81 m/s²), initial position and velocity vectors, and coefficient of restitution for potential bounces. The model deliberately excludes spin effects and complex aerodynamics to maintain computational efficiency and implementation simplicity across both simulation and real-world environments.

**Trajectory Prediction**    The prediction process is implemented using the following Algorithm 1. This algorithm applies the ballistic equations to project the ball's trajectory forward in time and determine exactly when and where it will intersect the hitting plane. The computed hittable position, impact velocity, and arrival time are then passed to the robot control system for interception planning.

The prediction algorithm includes special handling for various scenarios: trajectories that never intersect the hitting plane, balls moving away from the hitting plane, multiple potential intersections (selecting the earliest valid one), and bounces off other surfaces before reaching the hitting plane.

**Integration with Robot Control**    The predicted hittable position, impact velocity, and arrival time are continuously updated and provided to the robot control system, enabling it to plan and execute appropriate interception movements. This real-time prediction allows the robot to adjust its position and orientation to successfully hit the ball even when the ball follows an unexpected trajectory.

---

**Algorithm 1** Predict Hittable Position

---

**Require:** $p_0$ (initial position), $v_0$ (initial velocity), $t_{current}$ (current time), $y_{plane}$ (hitting plane y-coordinate)
**Ensure:** Returns hit position, velocity and arrival time, or null if not hittable
 1: $g \leftarrow (0, 0, -9.81)$ Gravity vector
 2: **if** $|v_0.y| < \epsilon$ **then**
 3:    Ball moving parallel to hitting plane
 4:    **return** null
 5: **end if**
 6: $t_{hit} \leftarrow (y_{plane} - p_0.y)/v_0.y$
 7: **if** $t_{hit} < 0$ **then**
 8:    Ball already passed the plane
 9:    **return** null
10: **end if**
11: $p_{hit} \leftarrow p_0 + v_0 \cdot t_{hit} + \frac{1}{2}g \cdot t_{hit}^2$
12: $p_{hit}.y \leftarrow y_{plane}$ Ensure exact y-coordinate
13: $v_{hit} \leftarrow v_0 + g \cdot t_{hit}$
14: $t_{arrival} \leftarrow t_{current} + t_{hit}$
15: **return** $(p_{hit}, v_{hit}, t_{arrival})$

---

## D  DATASET DESCRIPTION

This section details the two critical datasets developed for our table tennis robot system. These datasets support the system's two core components: the Fast-Slow System, focused on precise paddle-contact control, and the IMPACT model (Imitation-based Motion Planning And Control Technology), focused on effective hitting strategies. Below, we describe the collection processes, annotation methods, and how these datasets provide the foundation for the system's performance.

### D.1  DATASET FOR FAST-SLOW SYSTEM

#### D.1.1  DATA COLLECTION PROCESS

Our data collection process involves the following steps: The ball launcher randomly serves balls within a predetermined range. Based on the Ball Detection and Trajectory Prediction framework described above, we use the Intel RealSense D455 camera to record trajectory information, including position and velocity vectors. Once a hittable position is predicted, this coordinate is transformed into the robotic arm's base coordinate system. Using PyBullet, we compute the inverse kinematics to determine the joint values required to move to this position, which are then sent to the robotic arm via an EGM Controller. Leveraging the high frame rate capability of the Spike camera, we capture images of the exact moment of contact between the ball and the paddle. This process is repeated multiple times to build a comprehensive dataset.

#### D.1.2  DATA ANNOTATION PROCESS

**Pixel-to-Real-World Conversion**  A critical step in our data annotation process was establishing an accurate conversion ratio between pixel measurements in images and real-world dimensions. We developed a systematic approach using the known dimensions of the table tennis paddle as a reference.

The process involves the following steps:

1. **Image Acquisition**: We captured multiple high-resolution images of the table tennis paddle using the Spike camera.

2. **Image Preprocessing**: Each image undergoes preprocessing to reduce noise and enhance contrast, facilitating more accurate edge detection. We apply a center-crop operation to focus on the region containing the paddle.

3. **Paddle Detection**: Using color-based segmentation, we isolate the paddle from the background by defining a target color range (RGB: 65, 31, 31) with an appropriate tolerance value. This creates a binary mask representing the paddle area.

4. **Morphological Operations**: To refine the mask, we apply morphological operations including opening (to remove small noise artifacts) and closing (to fill small holes), using a 5×5 kernel.

5. **Connected Component Analysis**: We identify the largest connected component in the mask, which corresponds to the paddle, and filter out smaller noise components.

6. **Minimum Area Rectangle Fitting**: For the detected paddle region, we compute the minimum area rectangle that encloses the paddle contour, providing us with the paddle's pixel dimensions (width and height).

7. **Conversion Ratio Calculation**: Knowing the actual paddle dimensions (150mm × 150mm), we calculate the conversion ratio for both width and height:

$$\text{mm\_per\_pixel\_width} = \frac{\text{real\_width\_mm}}{\text{width\_pixels}} \tag{8}$$

$$\text{mm\_per\_pixel\_height} = \frac{\text{real\_height\_mm}}{\text{height\_pixels}} \tag{9}$$

8. **Average Conversion Ratio**: To improve accuracy, we average the width and height conversion ratios:

$$\text{mm\_per\_pixel\_avg} = \frac{\text{mm\_per\_pixel\_width} + \text{mm\_per\_pixel\_height}}{2} \tag{10}$$

9. **Multiple Image Processing**: We repeat this process across multiple images and compute the overall average conversion ratio to minimize measurement errors.

This methodology yielded a reliable pixel-to-millimeter conversion factor that was subsequently used throughout our data annotation pipeline to transform pixel coordinates in images to real-world spatial coordinates.

### D.1.3   Ball-Paddle Contact Detection

After establishing the pixel-to-millimeter conversion ratio, we developed a comprehensive algorithm to detect and analyze the contact between the table tennis ball and paddle in high-speed scenarios. This process is particularly challenging due to the rapid nature of the contact event, which typically occurs within milliseconds.

Our detection pipeline consists of the following key components:

1. **High Temporal Resolution Acquisition**: The Spike camera captures the ball-paddle interaction with exceptional temporal precision (microsecond-level), providing detailed information about the contact dynamics that conventional cameras would miss. The neuromorphic vision sensor outputs asynchronous spike signals rather than traditional frame-based images, enabling ultra-high temporal resolution.

2. **Ball Detection Strategy**: We implemented a dual-approach ball detection method:

   - **Hole-based Detection**: The primary method identifies the ball as a hole or void within the paddle region during contact. This approach is particularly effective when the ball partially occludes the paddle.
   - **Color-based Detection**: As a complementary approach, we detect the ball using a predefined color range (RGB: 79, 58, 34) with appropriate tolerance values, creating a binary mask for potential ball regions.

3. **Circularity Filtering**: To distinguish the ball from other objects or noise, we apply a circularity measure to each detected region:

$$\text{Circularity} = \frac{4\pi \times \text{Area}}{\text{Perimeter}^2} \tag{11}$$

Regions with circularity above 0.6 are considered potential ball candidates, as table tennis balls maintain their circular appearance even during high-speed motion.

4. **Proximity Analysis**: We prioritize detected ball regions that are within a 20-pixel radius of the paddle's edge, as these are most likely to represent actual contact points.

5. **Temporal Sequence Analysis**: By analyzing the sequential spike signals from the neuromorphic camera, we track the ball's trajectory before, during, and after contact with the paddle. This allows us to determine the exact moment of impact with microsecond precision.

6. **Coordinate System Transformation**: After detecting both the paddle and ball, we establish a paddle-centered coordinate system:

   - Origin: Center of the paddle
   - X-axis: Horizontal direction (positive rightward)
   - Y-axis: Vertical direction (positive upward)

7. **Contact Point Calculation**: The ball's position is transformed from pixel coordinates to this paddle-centered coordinate system and then converted to physical units (millimeters) using our established conversion ratio:

$$x_{mm} = (x_{ball} - x_{paddle}) \times mm\_per\_pixel \tag{12}$$

$$y_{mm} = (y_{paddle} - y_{ball}) \times mm\_per\_pixel \tag{13}$$

This ball-paddle contact detection methodology provides unprecedented insights into the dynamics of table tennis interactions. By leveraging the unique capabilities of neuromorphic vision sensors, we can capture and analyze high-speed interactions that would be impossible to observe with conventional imaging systems. The resulting data enables quantitative analysis of contact timing, location, and dynamics, which can be valuable for both sports science research and athlete training applications.

## D.2 DATASET FOR IMPACT

### D.2.1 DATA COLLECTION PROCESS

After successfully training the Fast-Slow System to ensure precise ball-paddle contact near the center of the paddle, we extended our data collection to include the hitting phase. Building upon our established interception capabilities, we implemented a structured process to collect data on effective hitting techniques. For the hitting phase data collection, we augmented our previous methodology with the following approach:

1. **Randomized Joint Angle Sampling**: To generate diverse hitting patterns, we implemented controlled random sampling on three critical joint axes:

   - Axis 3 (shoulder joint): Base angle of 15.0 degrees with random variation of ±10.0 degrees
   - Axis 5 (elbow joint): Base angle of 60.0 degrees with random variation of ±10.0 degrees
   - Axis 6 (wrist joint): Base angle of 0 degrees with random variation of ±20.0 degrees

2. **Hitting Execution**: For each trial, the robotic arm would:

   - First, intercept the ball using the Fast-Slow System's prediction
   - Apply the randomly generated joint angles at the moment of contact
   - Execute the hitting motion to return the ball to the opponent's side

3. **Outcome Recording**: We recorded whether the ball successfully landed on the opponent's side of the table, along with the precise landing location.

This approach allowed us to collect data on effective hitting strategies while leveraging our previously established ball interception capabilities. By systematically varying the joint angles during the hitting phase, we were able to explore a wide range of possible returns, creating a comprehensive dataset that captures the relationship between joint movements and resulting ball trajectories.

### D.2.2 DATA ANNOTATION PROCESS

For data annotation, we used an Intel RealSense L515 camera to record the landing position of the ball. We divided the opponent's side of the table into four quadrants (labeled A, B, C, and D) and encoded these landing zones using one-hot encoding. This encoded landing position information was incorporated as part of the input data, while the randomly sampled joint angles were used as labels. Using this structured dataset, we employed imitation learning techniques to train our IMPACT model, enabling it to learn the relationship between desired landing positions and the required joint movements to achieve them.

## E TRAINING DETAILS.

Both System 2 and IMPACT modules were trained for 2000 epochs using the Adam optimizer with an initial learning rate of 1e-3 and cosine annealing schedule. System 2 used a batch size of 32 with K=10 consecutive frames as input for trajectory prediction. The IMPACT module employed a batch size of 4 to handle the complexity of return planning tasks. All positional inputs were normalized to the range [0,1], while standard scaling was applied to velocity measurements to ensure numerical stability across varying ball conditions. Training was conducted on a workstation equipped with an NVIDIA RTX 4090 GPU.

## F ADDITIONAL REAL-WORLD EXPERIMENTS

### F.1 ULTRA-HIGH PRECISION EVALUATION

To further validate our system's precision capabilities and address questions regarding our target zone selection, we conducted additional experiments using 10cm radius targets following the methodology of Büchler et al. (Buchler et al., 2022).

**Experimental Setup.** We evaluated our system using 10cm radius target zones across the same four regions (A, B, C, D) used in our main experiments. This ultra-high precision threshold represents only 1.5% of the reachable table area, providing an extremely challenging benchmark for precision assessment.

Table 8: Ultra-high precision evaluation results with 10cm radius targets

| Method | A(%) | B(%) | C(%) | D(%) | AVG.(%) |
|---|---|---|---|---|---|
| HYSR (Buchler et al., 2022) | - | - | - | - | 8 |
| SpikePingpong | 31±4 | 32±3 | 29±3 | 35±2 | 31±3 |

**Results and Analysis.** Table 8 presents the results of our ultra-high precision evaluation. Our system achieves an average success rate of 31±3% within the 10cm target zones, substantially outperforming the 8% success rate reported by Büchler et al. (Buchler et al., 2022) using similar target specifications.

**Discussion.** These results validate our choice of 20cm and 30cm target zones as meaningful precision benchmarks while demonstrating that our system maintains reasonable performance even at ultra-high precision levels. The progressive degradation from 93% (30cm) to 70% (20cm) to 31% (10cm) reflects the inherent challenges of precise ball placement in table tennis, where even human players average only 53% success in 30cm zones. Our system's ability to achieve 31% success in 10cm targets represents a significant advancement in robotic table tennis precision control.

## G FAILURE CASE ANALYSIS

To gain deeper insights into our system's limitations and identify areas for improvement, we conducted a comprehensive failure analysis by categorizing all unsuccessful attempts during our precision evaluation experiments.

Table 9: Distribution of failure types in precision targeting experiments

| Failure Type | Percentage |
|---|---|
| Ball fails to cross net | 4.6% |
| Correct quadrant, outside 30cm circle | 79.1% |
| Wrong target quadrant | 12.5% |
| Ball fails to land on table | 3.8% |

**Results.**  Table 9 presents the distribution of failure types observed in our experiments. The analysis reveals that the majority of failures (79.1%) are near-misses where the system correctly identifies the target region but falls short of the required precision threshold.

**Analysis and Implications.**  The failure distribution indicates robust basic control capabilities with specific areas for improvement:

- **Near-miss failures (79.1%):** These cases demonstrate that our system successfully executes the fundamental task of directing the ball toward the intended table region but requires enhanced precision in fine-grained control. This suggests that improvements in trajectory optimization and control parameter tuning could yield significant performance gains.

- **Strategic errors (12.5%):** Wrong quadrant targeting indicates occasional failures in high-level decision making, potentially due to perception errors or planning inconsistencies under challenging conditions.

- **Fundamental control failures (8.4%):** The combined percentage of balls failing to cross the net or land on the table represents basic execution errors, suggesting room for improvement in fundamental trajectory planning and power control.

**Root Cause Analysis of Near-Misses.**  A deeper causal analysis reveals that the prevalence of "near-misses" stems from two primary root causes:

- **Unmodeled Spin Dynamics:** The most significant challenge is unmodeled ball spin. Extreme spin introduces complex aerodynamic effects (the Magnus effect) and alters bounce characteristics in ways our physics-based model does not capture. This can create small but critical, centimeter-level residual errors in our final trajectory prediction, often causing the ball to land just outside the target zone.

- **High Spatio-Temporal Sensitivity:** The task exhibits extreme sensitivity to the precise timing and location of the ball-paddle contact. Even minor variations in the robot's arrival time at the contact point can subtly alter the impact dynamics and the resulting ball trajectory. This high sensitivity means that even when the overall strategy is correct, slight execution imprecision can lead to a near-miss.

This analysis confirms that our system demonstrates robust basic control with the primary limitation being precision refinement rather than fundamental control failures, providing clear direction for future system enhancements.

## H  LIMITATIONS AND FUTURE WORK

### H.1  CURRENT LIMITATIONS

Despite the achievements demonstrated in this work, our *SpikePingpong* system has several limitations that present opportunities for future enhancement:

**Ball Spin Modeling:** Our current system does not account for ball spin, which significantly affects optimal interception strategies for different spin types. The Magnus effect and varying bounce characteristics of topspin, backspin, and sidespin balls can lead to trajectory deviations that our physics-based models do not capture.

**Human Player Adaptability:** Performance against human players remains challenging due to the complex and unpredictable trajectories they generate compared to our controlled training conditions. Human players exhibit diverse playing styles, strategic variations, and adaptive behaviors that exceed the scope of our current training data.

## H.2 FUTURE RESEARCH DIRECTIONS

Building upon the foundation established by *SpikePingpong*, we identify several promising directions for future research:

**Spin Dynamics Integration:** Incorporating comprehensive spin modeling into both the Fast-Slow system architecture and IMPACT module to handle the full spectrum of ball spin effects on trajectory prediction and strategic planning.

**Adaptive Learning Framework:** Enhancing the system's adaptability to diverse playing styles through online learning mechanisms that can adjust to opponent strategies and playing patterns in real-time.

**Strategic Gameplay Planning:** Developing advanced strategic planning capabilities for human-robot interaction, including opponent modeling, tactical sequence planning, and adaptive game strategy formulation.

## I   USE OF LARGE LANGUAGE MODELS

Large Language Models (LLMs) were used in a limited capacity during the preparation of this manuscript. Specifically, LLMs were employed solely for:

- Grammar correction and proofreading
- Sentence structure improvement and clarity enhancement
- Minor stylistic refinements to improve readability

LLMs were not involved in research ideation, experimental design, data analysis, or the generation of scientific content. All technical contributions, methodological innovations, experimental results, and scientific insights presented in this work are entirely the product of the authors' original research.

