# OpenReview forum: "SpikePingpong: Spike Vision-based Fast-Slow Pingpong Robot System"
_ICLR.cc/2026/Conference — ICLR 2026 Poster_

### Official Review · Reviewer_Tf5j · 2025-10-30

**Soundness:** 3
**Presentation:** 3
**Contribution:** 1
**Rating:** 0
**Confidence:** 4

**Summary:**

This paper is about a robotic system and related algorithms/models for playing pingpong using a robot manipulator. The system is based in the fast-slow principle by Kahneman, with a fast system that does rapid ball detection and physics-based prediction, and a slow system that improves the fast system with a spike-based calibration, then both systems are combined with a form of supervised imitation learning called IMPACT.

The contributions are:
-a comprehensive robotic table tennis system for high-speed manipulation via a cognitive-inspired architecture
- A fast-slow perception system combining imitation learning with accurate trajectory prediction.
- Extensive experiments showing that the proposed system improves the task to 92% success rate in 30cm zones.

**Strengths:**

- The paper is very well written and clear.
- I believe this is an interesting robotics problem, with some machine learning usage, definitely in the application domain of robotics this is an improvement over the state of the art and shows the power of learning systems.
- I believe the evaluation is correct, there are some baselines (not many choices in robotics), and there are good ablation results showing the contribution and necessity of the different components in the proposed method.

**Weaknesses:**

- My major complaing about this paper is that its contributions are in the robotics domain, and not in the machine learning domain. To me from the paper it is not clear what the machine learning community can learn from this paper, and while I think it is a great robotics paper, it does not seem to fix into a machine learning conference. Possibly this paper can be accepted by CORL or any other robotics conference or a journal.
- I believe the claim about a fast-slow system with two components that are working at different temporal frequency is a bit of a stretch, in terms that the system does not have cognition, so its not a cognitive architecture, it is just imitation of a well known cognitive principle (system 1 and system 2).
- There are no relevant machine learning model training details, for example there is no mention of data splits, cross-validation, or standard machine learning setup, and I believe this is because the paper focuses on the robotics concepts. I am sure the authors trained models properly and there was a validation set to check for overfitting, but this is not described in the paper so a future reader cannot check for generalization.

**Questions:**

- What is the machine learning contribution in this paper? What would machine learning people at ICLR learn from reading this paper?

---

> ### Author Response · Authors · 2025-11-20
> **Response to Reviewer Tf5j (part 1)**
>
> Dear Reviewer Tf5j,
>
> We sincerely thank you for your detailed and constructive review of our paper.  We will address your specific points below.
>
> **Q1: Concerns that the paper is a pure robotics engineering work with limited contribution to the machine learning community.**
>
> A1: We would like to clarify that our work targets a fundamental gap in the current literature: **high-speed decision-making under extreme temporal constraints**.
>
> Most recent ML breakthroughs target static or quasi-static domains (e.g., text, images), where latency and closed-loop dynamics are secondary. In contrast, physical-world systems must act under strict time budgets and rapidly changing dynamics, where conventional offline or high-latency methods break down. We utilize table tennis as a rigorous testbed to validate a novel framework for designing ML policies that remain reliable under millisecond-level latency and noisy real-world sensing.
>
> Specifically, we distill our technical advancements into two fundamental contributions to the Machine Learning community:
>
> - **A Neuro-Symbolic Framework for Asynchronous Residual Learning:**
>
>   The core ML challenge in high-speed dynamics is the latency-precision trade-off: physics models are fast but biased (ignoring aerodynamics), while deep networks are precise but slow and data-hungry.
>
>   - **The Contribution:** We propose a Physics-Regularized Residual Learning architecture that fuses heterogeneous modalities. It integrates explicit physical priors (System 1) with high-frequency neuromorphic spike streams (System 2) via a Transformer-based residual estimator.
>   - **Why it matters:** This introduces a generalizable methodology for multi-modal neuro-symbolic integration. Unlike standard fusion which treats inputs equally, our model uses asynchronous spike data specifically to learn the non-linear error manifold of the physical model. This effectively bridges the gap between symbolic reasoning (physics) and continuous representation learning (neural networks), solving the temporal alignment problem in processing asynchronous event data.
>
> - **Precision-Aware Conditional Policy Learning via Explicit State Decomposition:**
>
>   End-to-end visuomotor policies (mapping pixels directly to actions) often struggle with the precision required for high-speed interception, as they must simultaneously learn visual representation and motor control.
>
>   - **The Contribution:** We propose a **state-decomposed imitation learning framework** (IMPACT). By explicitly decoupling high-frequency trajectory estimation (provided by our Fast-Slow system) from motion planning, we formulate the striking task as a **conditional inverse dynamics problem**. The IMPACT module utilizes a Transformer to learn a precise mapping from *[Ball State, Target Token]* to *[Robot Joint Adjustments]*.
>
>   - **Why it matters:** This approach demonstrates the efficacy of **Structured Imitation Learning**. By injecting structure (explicit state estimation) into the learning pipeline, we convert a complex POMDP (Partially Observable Markov Decision Process) into a more tractable supervised learning problem. This offers a clear path for the ML community to solve high-precision manipulation tasks where pure reinforcement learning or end-to-end imitation is sample-inefficient or unstable.
>
> Beyond table tennis, this paradigm of physics-regularized perception coupled with structured policy decomposition offers a scalable blueprint for general embodied intelligence. For instance, in **autonomous driving**, the same principle applies: a fast reflexive controller handles immediate stability, while a slower vision model continuously injects strategic guidance (e.g., lane planning) into the control loop, optimizing performance without sacrificing safety or reaction speed.
>
> In conclusion, we specifically submitted to ICLR under the primary area of **"applications to robotics, autonomy, and planning,"** a subject explicitly welcomed in the conference's Call for Papers.
>
> ---
>
> **Q2: Concerns that labeling the architecture as a "Fast-Slow system" is a stretch, as the robot lacks actual cognition.**
>
> A2: The term "cognitive-inspired" was intended to describe our functional emulation of the human decision process (rapid judgment followed by refined adjustment), rather than to claim cognitive capabilities.
>
> However, to avoid any potential confusion or overstatement, we have removed the "cognitive-inspired" terminology from the revised manuscript.

---

> > ### Author Response · Authors · 2025-11-20
> > **Response to Reviewer Tf5j (part 2)**
> >
> > **Q3: Absence of detailed machine learning training protocols, specifically data splits and validation procedures to ensure generalization.**
> >
> > A3: We would like to clarify that we did provide a summary of our training setup in Appendix E ("Training Details"), which covers our choice of optimizer, learning rate schedule, batch sizes, and data normalization techniques.
> >
> > To further address the reviewer's valid concerns about reproducibility and generalization, we have expanded Section 4.1 to explicitly detail our ML evaluation protocol:
> >
> > - **Standard ML Validation (Offline):**
> >   For the ball-racket contact precision metrics (Table 1), we strictly followed standard ML practices. We collected a dataset of 1,000 trajectories and employed a standard 80/10/10 split (train/validation/test). The reported metrics are on the unseen test set, ensuring that our residual learning module (System 2) does not overfit to memorized trajectories.
> >
> > - **Stochastic Real-World Evaluation (Online):**
> >   The success rates (Tables 2 & 3) were evaluated online in the physical world. Crucially, these were not replay tests. For each trial, the ball launcher generated completely randomized trajectories with varying spin, speed, and initial coordinates. This ensures that the test distribution is stochastic and distinct from the fixed samples seen during training.
> >
> > - **Strong Out-of-Distribution (OOD) Generalization:**
> >   To rigorously prove that our model has learned robust representations rather than overfitting, we conducted two additional challenging OOD experiments:
> >   - Unseen Launcher Configurations: We physically relocated the launcher 30 cm off-center (left and right), creating incident angles never seen during training. The system maintained a high success rate of 74% (76% left / 72% right), demonstrating robust generalization to novel spatial distributions.
> >   - Zero-Shot Unseen Human Opponent: While our training data involved a single human demonstrator, we tested the system zero-shot against a completely new player (Person B). Despite the unstructured variability of human throws, the system achieved a 31% success rate. This confirms valid transfer learning to novel ball dynamics without any fine-tuning.
> >
> > We have updated the manuscript to highlight these rigorous evaluation protocols, ensuring future readers can clearly assess the model's generalization capabilities.

---

> ### Author Response · Authors · 2025-11-24
> **Follow-up on Our Response to Your Review**
>
> Dear Reviewer Tf5j,
>
> We hope this message finds you well.
>
> As the discussion period is progressing, we wanted to kindly check if you have had a chance to review our detailed response to your comments. We believe we have addressed your major concerns regarding the paper's positioning and evaluation:
>
> 1.  **ML Contribution:** We have clarified how our **Neuro-Symbolic Residual Learning** and **Structured Imitation Learning** frameworks contribute specifically to the ML community by solving high-speed latency-precision trade-offs.
> 2.  **Terminology:** We have removed the "cognitive-inspired" terminology to prevent overstatement.
> 3.  **Generalization & Protocols:** We have provided the exact training splits (80/10/10) and highlighted our **Out-of-Distribution (OOD)** experiments (unseen launcher positions and human opponents) to demonstrate robust generalization.
>
> We would greatly appreciate it if you could let us know if these clarifications and revisions satisfactorily address your concerns. We are happy to provide further details if needed.
>
> Best regards,
>
> The Authors

---

> ### Author Response · Authors · 2025-11-26
> **Follow-up on Our Response to Your Review**
>
> Dear Reviewer Tf5j,
>
> As the discussion period is ending very soon, we wanted to gently check if you have had a chance to consider our previous responses.
>
> We believe we have effectively addressed your concerns regarding the machine learning contributions and experimental settings. Since your feedback is crucial to the final decision of our paper, we would deeply appreciate it if you could let us know if our clarifications are satisfactory or if you have any remaining questions.
>
> Thank you for your time.
>
> Best regards,
>
> The Authors

---

> ### Author Response · Authors · 2025-11-28
> **Follow-up on Our Response to Your Review**
>
> Dear Reviewer Tf5j,
>
> As the discussion period is concluding shortly, we are writing to kindly ask if you have had a chance to review our detailed responses.
>
> We have made significant efforts to specifically address your concerns regarding the Machine Learning contributions (our Neuro-Symbolic framework and Structured Imitation Learning) and the experimental rigor (including the explicit training/test splits and OOD generalization tests).
>
> We would greatly value your feedback on whether these clarifications and revisions have successfully resolved your concerns regarding the paper's fit and evaluation.
>
> Best regards,
>
> The Authors

---

### Official Review · Reviewer_RtEN · 2025-11-03

**Soundness:** 3
**Presentation:** 3
**Contribution:** 3
**Rating:** 8
**Confidence:** 4

**Summary:**

This paper presents SpikePingpong, a robotic table tennis system that integrates spike-based vision with imitation learning. The system employs a cognitive-inspired Fast-Slow architecture where System 1 provides rapid ball detection and physics-based trajectory prediction, while System 2 uses spike camera data for refined trajectory corrections. The IMPACT module handles strategic ball striking through imitation learning. The system achieves high success rate in the real world.

**Strengths:**

- The Fast-Slow system architecture is well-motivated and technically sound, combining physics-based modeling with learned corrections.
- The spike camera integration for capturing millisecond-level ball-paddle contact is innovative and well-executed.
- The experimental validation includes solid real-world deployment and comprehensive comparisons and ablations.

**Weaknesses:**

- The task focuses solely on table tennis. There are many task-specific design choices with insufficient analysis of what methodology generalizes beyond table tennis (and applies to more general dynamic manipulation tasks).
- There lacks a in-depth failure analysis. Table 8 shows distributions but doesn't explain root causes of 79.1% near-miss failures or how failures vary by ball speed, spin, trajectory.
- Minor: Wrong citation in Table 3: "Diffusion Policy (Zhao et al., 2023)" should be "(Chi et al., 2023)".

**Questions:**

- Why is the reported diffusion policy inference time so slow (2437.22ms)? If using DDIM during inference with around 10 denoising steps, the inference should be around 100ms if using image input, and should be even faster if just using state input.
- Do ACT and diffusion policy baselines implementations have exact same state-based inputs/outputs as IMPACT or do they use image inputs?
- Previous work like UMI [1] is able to do dynamic tasks like tossing with asynchronous inference with latency matching using diffusion policy. How is policy inference implemented for baseline policies? Was asynchronous inference with latency matching attempted as well?

[1] Universal Manipulation Interface: In-The-Wild Robot Teaching Without In-The-Wild Robots. Cheng Chi, Zhenjia Xu, Chuer Pan, Eric Cousineau, Benjamin Burchfiel, Siyuan Feng, Russ Tedrake, Shuran Song.

---

> ### Author Response · Authors · 2025-11-20
> **Response to Reviewer RtEN (part 1)**
>
> Dear Reviewer RtEN,
>
> We sincerely thank you for your detailed and constructive review of our paper.  We will address your specific points below.
>
> **Q1: Concerns about the task-specific nature of table tennis and the limited generalizability of the methodology to other dynamic manipulation tasks.**
>
> A1: We chose table tennis precisely because it represents the **upper bound of difficulty** in dynamic manipulation. The ball is small (40mm), light (2.7g), and moves at extremely high speeds with complex spin dynamics, demanding far greater temporal resolution and control precision than typical tasks like catching thrown objects.
>
> Our framework is designed to be generalizable to other high-speed interception tasks. The Fast-Slow perception architecture and the IMPACT control module are task-agnostic principles. For example, in catching a falling glass, System 1 could rapidly guide the hand to the object's predicted fall line, while System 2 continuously refines the precise grasp timing and finger positioning based on high-frequency visual feedback, ensuring a secure catch despite air resistance or tumbling motion.
>
> ---
>
> **Q2: Lack of in-depth failure analysis, particularly regarding the root causes of near-misses and their correlation with ball speed, spin, and trajectory.**
> A2: We attribute the "near-misses" to two primary factors:
>
> - **For unmodeled spin dynamics:** Extreme spin (especially sidespin) introduces aerodynamic forces that are not fully captured by the coarse physics model, causing centimeter-level deviations in the final landing point.
> - **For spatio-temporal sensitivity:** At higher ball speeds, the tolerance for synchronization tightens to milliseconds. Minor temporal mismatches in robot actuation can subtly alter the rebound vector.
>
> Consequently, failure rates are observed to **increase with higher ball speeds and extreme spins,** as these conditions amplify both aerodynamic unpredictability and the demand for temporal precision. We have expanded the manuscript with this detailed correlation analysis.
>
> ---
>
> **Q3: Wrong citation in Table 3.**
>
> A3: We will correct the reference for Diffusion Policy to "(Chi et al., 2023)" in Table 2 and throughout the final manuscript.
>
> ---
>
> **Q4: Inquiry about the unexpectedly high inference latency reported for the Diffusion Policy and suggestion to use faster samplers like DDIM.**
>
> A4: The reported latency was for DDPM (1000 steps). We re-evaluated using DDIM (10 steps), reducing latency to 25.18ms. However, even this delay compromises the perception window and overall success rate. We have updated Tables 2, 3, and 6 in the revised manuscript to reflect these optimized baseline results and their impact.
>
> | Method            |          Inference time (ms)          |
> | :---------------- | :-----------------------------------: |
> | Diffusion Policy  | 25.18 |
> | ACT               |                 7.15                  |
> | **SpikePingpong** |               **0.407**               |
>
>
>
> | Method | A (30cm) | A (20cm) | B (30cm) | B (20cm) | C (30cm) | C (20cm) | D (30cm) | D (20cm) | **Avg. (30cm)** | **Avg. (20cm)** |
> | :--- | :---: | :---: | :---: | :---: | :---: | :---: | :---: | :---: | :---: | :---: |
> | Diffusion Policy (DDIM) | 3 ± 2 | 1 ± 1 | 4 ± 3 | 0 ± 0 | 2 ± 2 | 1 ± 1 | 3 ± 2 | 0 ± 0 | 3 ± 2 | 1 ± 1 |
> | **SpikePingpong (Ours)** | **91 ± 3** | **69 ± 4** | **93 ± 2** | **72 ± 3** | **92 ± 4** | **70 ± 5** | **93 ± 3** | **71 ± 4** | **92 ± 3** | **70 ± 4** |

---

> > ### Author Response · Authors · 2025-11-20
> > **Response to Reviewer RtEN (part 2)**
> >
> > **Q5: Clarification on whether ACT and Diffusion Policy baselines used image inputs or the same state-based inputs as IMPACT.**
> >
> > A5: Initially, we used image inputs to adhere to the original baseline papers. Following your suggestion for a fairer comparison, we re-trained both ACT and Diffusion Policy using **only state-based inputs** (identical to IMPACT).
> > Interestingly, both baselines improved without visual inputs, confirming that compact states are more effective for high-speed tasks. We have updated **Tables 2 and 3** with these stronger state-based results.
> >
> > | Method                        |  A (30cm)  |  A (20cm)  |  B (30cm)  |  B (20cm)  |  C (30cm)  |  C (20cm)  |  D (30cm)  |  D (20cm)  | **Avg. (30cm)** | **Avg. (20cm)** |
> > | :---------------------------- | :--------: | :--------: | :--------: | :--------: | :--------: | :--------: | :--------: | :--------: | :-------------: | :-------------: |
> > | Human Avg.                    |   48 ± 6   |   28 ± 4   |   52 ± 5   |   32 ± 3   |   56 ± 7   |   38 ± 5   |   54 ± 4   |   34 ± 6   |   **53 ± 6**    |   **33 ± 5**    |
> > | Diffusion Policy (w/ vision)  |   3 ± 2    |   1 ± 1    |   4 ± 3    |   0 ± 0    |   2 ± 2    |   1 ± 1    |   3 ± 2    |   0 ± 0    |    **3 ± 2**    |    **1 ± 1**    |
> > | Diffusion Policy (w/o vision) |   6 ± 3    |   2 ± 1    |   7 ± 4    |   2 ± 2    |   5 ± 2    |   1 ± 1    |   6 ± 3    |   1 ± 1    |    **6 ± 3**    |    **2 ± 1**    |
> > | ACT (w/ vision)               |   11 ± 4   |   4 ± 2    |   12 ± 5   |   4 ± 1    |   10 ± 3   |   2 ± 1    |   14 ± 4   |   5 ± 2    |   **12 ± 4**    |    **4 ± 2**    |
> > | ACT (w/o vision)              |   18 ± 5   |   7 ± 3    |   20 ± 6   |   8 ± 3    |   17 ± 4   |   6 ± 2    |   19 ± 5   |   7 ± 2    |   **19 ± 5**    |    **7 ± 3**    |
> > | **SpikePingpong (Ours)**      | **91 ± 3** | **69 ± 4** | **93 ± 2** | **72 ± 3** | **92 ± 4** | **70 ± 5** | **93 ± 3** | **71 ± 4** |   **92 ± 3**    |   **70 ± 4**    |
> >
> > ---
> >
> > **Q6: Inquiry about whether baseline policies used asynchronous inference (like UMI)**
> >
> > A6: We use a synchronous, "just-in-time" strategy: the system observes the ball until the calculated "latest decision point" and then executes a single, one-shot forward pass to generate the striking motion. This "last-moment" inference ensures the action is based on the most recent, high-quality data, which is critical for interception accuracy, unlike continuous asynchronous updates that introduce latency and struggle with rapid state evolution.

---

### Official Review · Reviewer_wtEY · 2025-11-04

**Soundness:** 3
**Presentation:** 3
**Contribution:** 3
**Rating:** 8
**Confidence:** 4

**Summary:**

The paper presents a learning-based robotic system for autonomous ping pong playing. It consists of three main modules: a perception module for detecting ping pong ball trajectory, a module for detecting hitting positions supervised with GT obtained from privileged spike cameras, and a policy module for predicting parameters for ball hitting. The system is learned with successful demonstrations obtained from an automated pipeline with random hitting configurations. It is shown to be able to condition on desired landing region, as well as outperforming human baseline in consecutive ball hitting returns.

**Strengths:**

- The paper is overall well-written and easy to follow.
- It is impressive to see the demo included in the supplementary submission, which demonstrates the robustness of the system.
- The experiments are comprehensive to demonstrate the importance of the design choices made in this work as well as the overall robustness of the system.
- As a system-oriented paper, the design choices and their details presented in the paper can be useful references for future works in similar direction.

**Weaknesses:**

- No major weaknesses.
- Typo in Figure 2: “inversive kinamatics”

**Questions:**

None.

---

> ### Author Response · Authors · 2025-11-20
> **Response to Reviewer wtEY**
>
> Dear Reviewer wtEY,
>
> We are sincerely grateful for your positive and encouraging review of our work. We deeply appreciate your assessment that our paper has "no major weaknesses," which is a great motivation for our team.
>
> **Q: Typo in Figure 2.**
>
> A: We have corrected "inversive kinamatics" to "inverse kinematics" in the revised manuscript. We appreciate your attention to detail, which helps us improve the quality of our paper.

---

### Official Review · Reviewer_Yh92 · 2025-11-07

**Soundness:** 3
**Presentation:** 3
**Contribution:** 3
**Rating:** 8
**Confidence:** 5

**Summary:**

This paper presents a vision-equipped robotic ping-pong system using an ABB IRB-120 arm with a standard table-tennis racket. The framework consists of three main components: (1) System 1, which provides ball detection from rgbd and trajectory prediction; (2) System 2, which refines the predicted hittable position; and (3) an Imitation-based module (IMPACT) that learns optimal striking strategies from human demonstrations. Together, these modules form a closed-loop perception–planning–control system capable of perceiving, planning, and executing appropriate striking motions in response to the opponent’s incoming shots. Real-world experiments demonstrate the proposed method‘s effectiveness.

**Strengths:**

+ Well-structured and intuitive architecture. The modular pipeline from perception to control is clear and logically connected.

+ Detailed modular implementation. The paper provides comprehensive implementation details for each subsystem, enabling reproducibility and practical insights for future robotic applications.

+ Good real-world results. Experimental evaluation on the physical ABB IRB-120 setup demonstrates reliable tracking, accurate returns, and stable rally performance, confirming real-time feasibility.

**Weaknesses:**

The main weakness of this paper lies in its limited adaptation and generalization capability. Since the Stage-3 action generation (IMPACT) relies purely on imitation learning, the system lacks the ability to adapt to unseen or out-of-distribution ball trajectories, such as those with different spins, velocities, or bounce patterns. The evaluation also appears to be confined to in-domain scenarios, and it remains unclear whether the tests include unseen human/robot launch opponents or novel shot types. Moreover, the demonstration data are said to come from both a ball-launching machine and human players, but the paper does not specify which subsets were used for training and which for testing, nor does it clarify whether the demonstrator shown in the supplementary video is the same person who provided the training data. The robot’s striking behavior in the video further appears to be limited to only two distinct motion patterns, suggesting a lack of diversity and adaptability in the learned control policy. A deeper analysis of how the system handles different incoming trajectories and human play styles, given that variations in hitting points and trajectories across players can be substantial, would greatly strengthen the paper and highlight the system’s robustness beyond demonstration-specific conditions.

**Questions:**

see "weaknesses"

---

> ### Author Response · Authors · 2025-11-20
> **Response to Reviewer Yh92**
>
> Dear Reviewer Yh92,
>
> We sincerely thank you for your detailed and constructive review of our paper.  We will address your specific points below.
>
> **Q1: Concerns about the system's limited generalization to OOD trajectories and unseen scenarios.**
>
> A1: We have addressed the generalization concern by evaluating our system in two distinct out-of-distribution (OOD) scenarios:
>
> - **Unseen Launcher Positions:** We physically relocated the launcher 30 cm to the left and right of the training center. Despite these unseen incoming angles, the system achieved success rates of 76% and 72% respectively (avg. 74%).
> - **Unseen Human Opponents:** We tested the system zero-shot against a new human player (Person B). It achieved a 31% success rate, demonstrating valid transfer to completely novel ball dynamics.
>
> These results (added to Table 4 and Table 5) confirm that our policy maintains a commendable success rate even under significant distributional shifts.
>
> | Condition | Avg. Success Rate (30cm) | Avg. Success Rate (20cm) |
> | :--- | :---: | :---: |
> | **In-Distribution** | 92 ± 3% | 70 ± 4% |
> | **Out-of-Dist.** | 74 ± 5% | 52 ± 6% |
>
> | Precision | Condition | A (%) | B (%) | C (%) | D (%) | Avg. (%) |
> | :--- | :--- | :---: | :---: | :---: | :---: | :---: |
> | **30cm** | Seen Player (Person A) | 51±5 | 47±3 | 42±3 | 50±4 | **47±3** |
> | **30cm** | Unseen Player (Person B) | 35±6 | 31±5 | 27±5 | 32±4 | **31±5** |
> | **20cm** | Seen Player (Person A) | 29±2 | 23±4 | 27±2 | 28±3 | **27±3** |
> | **20cm** | Unseen Player (Person B) | 18±3 | 13±4 | 16±3 | 14±4 | **15±4** |
>
> ---
>
> **Q2: Lack of clarity regarding train/test data splits and whether the human demonstrator in the video was seen during training.**
>
> A2: We have clarified the data composition and experimental settings as follows:
> - **For the train and test data split,** we collected 2,000 trajectories for training. Crucially, testing was conducted online in the real world, where the launcher generated completely randomized trajectories (varying spin, speed, and location) for each trial, ensuring the test scenarios were stochastic and distinct from the training samples.
> - **For the human evaluation,** the demonstrator in the video is indeed the same person used for training. However, since human throws are inherently uncontrollable and stochastic, the test trajectories naturally differ from the training data. To further validate generalization, we added a new zero-shot experiment against a completely unseen human player.
>
> | Precision | Condition | A (%) | B (%) | C (%) | D (%) | Avg. (%) |
> | :--- | :--- | :---: | :---: | :---: | :---: | :---: |
> | **30cm** | Seen Player (Person A) | 51±5 | 47±3 | 42±3 | 50±4 | **47±3** |
> | **30cm** | Unseen Player (Person B) | 35±6 | 31±5 | 27±5 | 32±4 | **31±5** |
> | **20cm** | Seen Player (Person A) | 29±2 | 23±4 | 27±2 | 28±3 | **27±3** |
> | **20cm** | Unseen Player (Person B) | 18±3 | 13±4 | 16±3 | 14±4 | **15±4** |
>
>
> ---
>
> **Q3: Concerns about the perceived lack of motion diversity and adaptability in the robot’s striking behavior.**
>
> A3: The seemingly limited motion patterns reflect a deliberate **two-phase strategy** rather than a lack of adaptability:
>
> - **Interception:** The robot uses all 6 joints to reach the predicted hitting point, ensuring coverage of the entire workspace.
> - **Striking:** From this pose, the policy modulates 3 key joints to execute the return.
>
> Our validation (>1,000 trials) confirms that controlling these 3 joints is sufficient to target **95% of the opponent's table**, effectively balancing control simplicity with high-precision placement capabilities.

---

### Author Response · Authors · 2025-11-24
**Global Response [Update of PDF & Point-by-Point Reply]**

First and foremost, we would like to express our sincere gratitude to the Reviewers, Area Chairs, and Program Chairs for their time and dedicated effort. We have carefully studied **all comments** and have provided detailed **point-by-point** responses under each Reviewer's section.

We are encouraged not only by the positive scores but, more importantly, by the recognition of our work from the reviewers across three key dimensions:

*   **Method:** "Well-structured and intuitive architecture" (`Reviewer Yh92`); "can be useful references for future works in similar directions" (`Reviewer wtEY`); "well-motivated and technically sound" (`Reviewer RtEN`); "innovative and well-executed" (`Reviewer RtEN`); "shows the power of learning systems" (`Reviewer Tf5j`).
*   **Evaluation:** "Reliable tracking, accurate returns, and stable rally performance" (`Reviewer Yh92`); "comprehensive" (`Reviewer wtEY`); "solid real-world deployment" (`Reviewer RtEN`); "good ablation results" (`Reviewer Tf5j`).
*   **Presentation:** "Clear and logically connected" (`Reviewer Yh92`); "well-written and easy to follow" (`Reviewer wtEY`); "very well written and clear" (`Reviewer Tf5j`).

In response to the questions, we have added rigorous **out-of-distribution experiments** (unseen launcher positions and a new human opponent) to verify robustness. We also clarified our technical contribution as a **neuro-symbolic residual learning framework** and re-evaluated baselines under optimized settings (state-based inputs, DDIM) for fairer comparison.

These updates, along with refined terminology, are detailed in the revised PDF (**highlighted in blue**). A summary of these changes follows:

- **Section 4.1 (Experiment Setting):**
    - To address `Reviewer RtEN`'s questions, we have added specific implementation details of the baseline method, including model inputs and the policy inference process.
    - To address `Reviewer Yh92` and `Reviewer Tf5j`'s questions, we have added a description of the validation protocol, including dataset partitioning and In-Distribution (ID) / Out-of-Distribution (OOD) experimental settings.

- **Section 4.2 (Main Results):**
    - To address `Reviewer RtEN`'s questions, we have conducted additional experiments with different variants of the baseline method and corrected the inference time measurements.
    - To address `Reviewer Yh92`'s questions, we have added new generalization experiments, covering different ball machine positions and results from human-robot gameplay matches.


- **Appendix G (Failure Case Analysis):**
    - To address `Reviewer RtEN`'s questions, we have included an in-depth analysis of the reasons for failure.

- **Minor Corrections & Terminology:**
    - We corrected spelling errors in Figure 2 and citation errors in Table 3. Additionally, we refined the description of "cognitive-inspired" throughout the paper to prevent ambiguity.


In particular, **regarding Reviewer Tf5j's concern about the paper's alignment with ICLR**, we respectfully highlight that this work was submitted under the primary area of **"Applications to Robotics, Autonomy, and Planning,"** a subject explicitly welcomed in the conference's Call for Papers. Beyond the application, we introduce a generalizable neuro-symbolic residual learning framework designed to resolve fundamental latency-precision trade-offs in high-speed physical systems. We believe this methodology offers distinct value to the ML community by demonstrating how structured learning can outperform end-to-end approaches in dynamic, resource-constrained environments.

We sincerely hope to engage in constructive dialogue with reviewers as we believe this exchange is vital for improving the quality of our work.

Best regards,

Authors of Submission 16484

---

### Meta-Review · Area_Chair_Q21d · 2025-12-03

**Summary:**

Three reviewers (Yh92, wtEY, RtEN) gave uniformly positive ratings of 8/10, praising the well-structured architecture, comprehensive experiments, and solid real-world deployment. Reviewer Tf5j gave a strong reject (0/10), questioning the paper's fit for a machine learning conference, the "cognitive-inspired" terminology, and claiming insufficient training details and validation protocols.

**Reviewer Concerns:**

The authors comprehensively addressed all concerns. They clarified the paper was submitted under the explicitly invited "applications to robotics" area and articulated two concrete ML contributions: neuro-symbolic asynchronous residual learning and structured imitation learning via state decomposition. They removed the "cognitive-inspired" terminology, added explicit 80/10/10 train/validation/test splits, and conducted new out-of-distribution experiments. Additional baseline experiments with optimized settings, failure analysis, and minor corrections were provided for other reviewers.

All concerns from Yh92, wtEY, and RtEN were fully addressed.

**Reviewer Scores:**

Reviewers Yh92, wtEY, and RtEN would likely have maintained their scores of 8, as their concerns were directly addressed. Reviewer Tf5j's liekly would have increased score with the lense that the paper was in scope.

---

### Decision · Program_Chairs · 2026-01-26

Accept (Poster)